# FAuNO: Semi-Asynchronous Federated Reinforcement Learning Framework for Task Offloading in Edge Systems

## Abstract

Edge computing addresses the growing data demands of connected-device networks by placing computational resources closer to end users through decentralized infrastructures. This decentralization challenges traditional, fully centralized orchestration, which suffers from latency and resource bottlenecks. We present **FAuNO**—*Federated Asynchronous Network Orchestrator*—a buffered, asynchronous *federated reinforcement-learning* (FRL) framework for decentralized task offloading in edge systems. FAuNO adopts an actor–critic architecture in which local actors learn node-specific dynamics and peer interactions, while a federated critic aggregates experience across agents to encourage efficient cooperation and improve overall system performance. Experiments in the *PeersimGym* environment show that FAuNO consistently matches or exceeds heuristic and federated multi-agent RL baselines in reducing task loss and latency, underscoring its adaptability to dynamic edge-computing scenarios. [1]

## 1 Introduction

The growth of connected device networks, such as the Internet of Things (IoT), has led to a surge in data generation. Traditionally, Cloud Computing handled these computational demands, but increased network traffic and latency became apparent as these networks expanded Min et al. (2019). Edge Computing (EC) extends the cloud by bringing computational resources closer to end-users, addressing latency and traffic issues Varghese & Buyya (2018). Despite distinguishing between Mobile Edge Computing (MEC) and Fog computing, this paper treats them interchangeably, focusing on their goal of minimizing device-to-cloud distances Yu et al. (2020). The EC paradigm distributes computational resources, making centralized network orchestration inefficient. Centralization would require aggregating data at a single node, straining the network, and creating a single point of failure Baek & Kaddoum (2023). This highlights the value of decentralized orchestration, particularly through Task Offloading (TO). Optimal TO in such distributed environments involves managing multiple factors, including task latency, energy consumption, and task completion reliability Zhu et al. (2019). Traditional optimization methods often struggle to efficiently manage these complex systems, due to the dynamic, time-varying, and complex environments of Edge Systems Xu et al. (2018). Reinforcement Learning (RL) Baek & Kaddoum (2023); Zhu et al. (2019), is a powerful candidate and dominant approach to solving the TO problem. Specifically, Multi-Agent Reinforcement Learning (MARL) has been explored as a promising solution for decentralized orchestration in Edge Systems Baek & Kaddoum (2023); Gao et al. (2022). The ability of MARL agents to iteratively learn optimal strategies through simultaneous interaction with an environment makes them particularly suited for decentralized edge systems Lin et al. (2023); Zhang et al. (2023). Due to the nature of MARL, it is common to have some form of message exchange Zhang et al. (2018); Baek & Kaddoum (2023) between participants, as this reduces the uncertainty generated by having multiple agents interacting simultaneously, making it particularly suitable for Federated Learning (FL), which has recently gained academic interest as an efficient and distributed approach to agent cooperation in learning Consul et al. (2024). When FL is applied to MARL and the agents only have partial observability of the state, as is commonly the case in decentralized systems where obtaining information about all nodes comes at a premium, it creates a paradigm known as Vertical Federated Reinforcement Learning (VFRL) Qi

---

[1]Repository: `https://anonymous.4open.science/r/FAuNO-C976/README.md`

et al. (2021). The MARL problem is transformed from one in which agents focus solely on their own objectives into a global optimization problem that accounts for the collective objectives of the participants in the federation. FL also mitigates the strain on the network by avoiding the exchange of large amounts of information, since agents only need to periodically share their learned updates that are aggregated into a global unified model solving the global objective. This enables agents to benefit from each other's knowledge while minimizing communication overhead. However, conventional FL suffers when stragglers delay aggregation or drop updates, reducing training efficiency and wasting samples. This can be addressed by adopting a buffered semi-asynchronous strategy, in which faster nodes continue contributing updates without waiting, while slower nodes are still able to align with the evolving global critic. FAuNO adopts a buffered semi-asynchronous strategy, where faster nodes continue contributing updates without waiting, while slower nodes are still able to align with the evolving global critic. In this way, we extend Federated Buffering Nguyen et al. (2021) to reinforcement learning, enabling continuous local training without being bottlenecked by stragglers. We summarize the motivations and principal contributions of this work below.

**Motivations & Contributions**

- We address the **TO problem in edge systems** by framing it within a **Partially Observable Markov Game (POMG)**, enabling decentralized decision-making under partial observability.
- We introduce **FAuNO**, the first framework to integrate **buffered semi-asynchronous aggregation** with **actor–critic MARL (PPO)** in a federated setting for edge offloading. Our adaptation of Fed-Buff to reinforcement learning enables faster agents to contribute multiple updates without waiting for stragglers, improving sample efficiency under heterogeneous conditions. By federating only the **critic** while keeping **actors local**, FAuNO mitigates heterogeneity, respects partial observability, and supports fully decentralized execution. Through empirical evaluation, we show that FAuNO **outperforms or matches FRL and heuristic baselines** in terms of task completion time and task completion.
- We **extend the PeersimGym** environment to support **federated update exchanges** over the simulated network (details in annex 7). This extension simulates the communication of the updates affecting how and when updates are propagated and aggregated. As a result, the evaluation reflects the conditions of realistic edge systems.

**Background & Related Work**

TO involves transferring computations from constrained devices to more capable ones, addressing the *what*, *where*, *how*, and *when* of offloading Fahimullah et al. (2022). TO methods include vertical offloading to higher-tier systems Qiu et al. (2019), horizontal offloading among peers Baek et al. (2019), and hybrid approaches Baek & Kaddoum (2023). Offloading target selection may prioritize proximity Van Le & Tham (2018); Yu et al. (2020) or queue length Baek et al. (2019), or consider unrestricted selection, accounting for consequences of offload failures. Failures are affected by factors like latency Dai et al. (2022), resource capacity Van Le & Tham (2018), energy shortages, or others Peng & et al. (2022). This study focuses on *Binary TO* Hamdi et al. (2022) for indivisible tasks with horizontal and vertical offloading.

RL has been applied to TO in both single-agent and multi-agent settings. In the single-agent case, TO is commonly modeled as an MDP and solved with Q-learning in Fog networks Baek et al. (2019), DQN in ad-hoc mobile clouds Van Le & Tham (2018), DDPG for task dependencies Liu et al. (2023), SARSA variants for real-time MEC Alfakih et al. (2020), and DQN extensions for delay-sensitive tasks Liu et al. (2022). Bandit formulations have also been used to simplify binary offloading while optimizing latency and energy Zhu et al. (2019). In the multi-agent case, MARL methods address resource allocation and collaboration in heterogeneous, partially observable environments. For example, in Baek & Kaddoum (2020) TO in Multi-Fog systems is modeled as a Stochastic Game, and a Deep Recurrent Q-Network (DRQN) with Gated Recurrent Units is employed to handle partial state observations.

FRL has been explored for TO in Edge systems, emphasizing agent cooperation. However, most RL research in TO focuses on parallel RL, where agents act on independent environment replicas, not considering the uncertainty introduced by shared environments. In Li et al. (2023), a multi-TO algorithm is developed that uses a Double Deep Q-Network (DDQN) and K-Nearest neighbors to obtain local offloading schemes. The agents then participate in training a global algorithm using

Table 1: Comparison of RL-based Task Offloading approaches.

| Work | Multi-Agent | Federated | Actor-Critic | Partially Obs. | Shared Env. | Buffered Async. | OSS Env. |
|------|:-----------:|:---------:|:------------:|:--------------:|:-----------:|:---------------:|:--------:|
| Baek et al. (2020) Baek & Kaddoum (2020) | ✓ | | DRQN | ✓ | | | |
| Baek et al. (2022) Baek & Kaddoum (2023) | ✓ | ✓ | ✓ | ✓ | ✓ | | |
| Zang et al. (2022) Zang et al. (2022) | ✓ | ✓ | DQN | ✓ | ✓ | | |
| Li et al. (2023) Li et al. (2023) | ✓ | ✓ | DDQN | | | | |
| Peng et al. (2024) Peng et al. (2024) | | ✓ | Dueling DQN | ✓ | | | |
| **FAuNO (ours)** | ✓ | ✓ | ✓ | ✓ | ✓ | ✓ | ✓ |

a weighted federated averaging algorithm. A unary outlier detection technique is used to manage stragglers.

In Consul et al. (2024), a hierarchical FRL model is proposed for frame aggregation and offloading of Internet of Medical Things data, optimizing energy and latency by aggregating learned parameters from body-area devices to edge and central servers. In Chen & Liu (2022) an FRL-based joint TO and resource allocation algorithm to minimize energy consumption on the IoT devices in the Network, considering a delay threshold and limited resources is proposed. The considered approach uses DDPG locally and a FedAvg McMahan et al. (2017) based algorithm for the global solution. In Tang & Wong (2022), a binary TO algorithm for MEC systems is proposed, employing dueling and double DQN with LSTM to improve long-term cost estimation for delay-sensitive tasks. In Zang et al. (2022), a scenario with multiple agents in the same environment is considered, and FEDOR – a Federated DRL framework for TO and resource allocation to maximize task processing is proposed. In FEDOR, Edge users collaborate with base stations for decisions, and a global model is aggregated using FedAvg, with an adaptive learning rate improving convergence. Although FEDOR considers multiple agents in the same scenario, the decision-making depends on base stations for smoothing the offloading decisions of the multiple agents. In Baek & Kaddoum (2023), FLoadNet is proposed as a framework that combines local actor networks with a centralized critic, trained synchronously in a federated manner, to enable collaborative task offloading in Edge-Fog-Cloud systems. Their solution learns what information to share between nodes to enhance cooperation and their offloading scheme learns the optimal paths for tasks to take through a Software-defined Network. In the Industrial IoT(IIoT) setting with dependency-based tasks, Peng et al. (2024) propose SCOF that considers a Federated Duelling DQN, that is aggregated with a FedAvg-based approach and utilizes differential privacy (DP) to improve the security of the update exchanges. Focusing on selecting the best offloading targets from a pool of Edge Servers.

Lastly, none of the studied solutions uses an environment that facilitates the comparison of the proposed algorithms, which we do by training and benchmarking our solution in the PeersimGym environment Metelo et al. (2024). The comparison with the related work is summarized in Table 1.

## 2 FEDERATED TASK OFFLOADING PROBLEM

In this section, we elaborate on the system modeling of our Edge System and formulate the TO problem as a global optimization problem that will be solved by all the participants in the network orchestration. Lastly, we formulate the local learning problem of the participants as a POMG.

### 2.1 SYSTEM MODEL

We consider a set of nodes $\mathcal{W} = W_1, \ldots, W_k$ comprising the network entities (e.g., edge servers, mobile users). These nodes offer computational resources to a set of clients, such as IoT sensors that require processing for collected data. Time is discretized into equidistant intervals $t \in \mathbb{N}_0$. The system includes two types of entities, as illustrated in Fig. 1. **Clients** generate computational workloads in the form of tasks for accessible nodes, following a Poisson process with rate $\lambda$; the set of all clients is denoted by $\dot{C}$. **Workers**, denoted by $\dot{W}$, provide computational resources and are represented as nodes with specific properties. Each worker $W^n$ maintains a task queue $Q_t^n$ at time $t$, with a maximum capacity $Q_{\max}^n$. Whenever the capacity is reached, no new tasks are accepted until there is available space. The Workers are characterized by the number of CPU cores $N_\phi^n$, the per-core frequency $\phi^n$ (in instructions per time step), and a transmission power budget $\mathcal{P}_n$. Workers periodically share their state with neighbors. A single machine can be both a worker and a client.

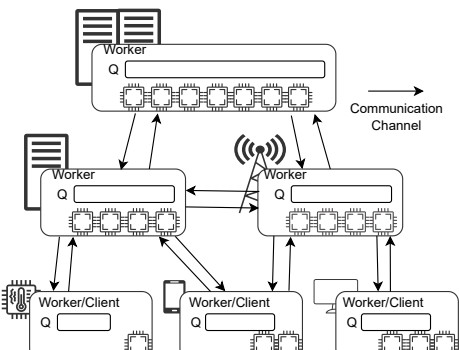

Figure 1: Edge System Architecture of our system model. The workers are capable of independently offloading tasks, exchanging information, and FL model updates through the communication channels.

**Task Model**: Computational requirements are modeled as tasks in our system. Let $T = \{\tau_i\}, i \in \mathbb{N}$ be the set of all tasks, where a task with ID $i$ is represented as $\tau_i = \langle i, \rho_i, \alpha_i^{\text{in}}, \alpha_i^{\text{out}}, \xi_i, \delta_i \rangle$, with the following attributes: $i$ as a unique task identifier, $\rho_i$ as the number of instructions to be processed, $\alpha_i^{\text{in}}$ as the total input data size, $\alpha_i^{\text{out}}$ as the output data size, $\xi_i$ as the CPU cycles per instruction, and $\delta_i$ as the task deadline, or maximum allowed latency for the return of results. A task is dropped if it arrives at a node with a full queue or if its deadline expires.

**Communication Model**: The communication model defines the latency of message transmission between nodes in the same neighborhood, where a node can only communicate directly with its neighbors. Each entity within a node can send and receive messages, modeled as the tuple $\langle \omega_i, \omega_j, \alpha \rangle$, where $\omega_i$ is the origin node ID, $\omega_j$ is the destination node ID, and $\alpha$ represents the message size. To measure transmission delay, we consider the *Shannon-Hartley* theorem Anttalainen (2003). According to this theorem, the latency for transmitting $\alpha$ bits between nodes $W_i$ and $W_j$ is given by:

$$T_{i,j}^{\text{comm}}(\alpha) = \frac{\alpha}{B_{i,j} \log_2(1 + 10^{\frac{\mathcal{P}_i + G_{i,j} - \omega_0}{10}})}, \tag{1}$$

where $T_{i,j}^{\text{comm}}(\alpha)$ is the transmission time, $B_{i,j}$ is the bandwidth between nodes, $\mathcal{P}_i$ is the source node's transmission power, $G_{i,j}$ is the channel gain, and $\omega_0$ is the noise power. See annex 10.1 for details on communication protocols.

## 2.2 PROBLEM FORMULATION

We aim to optimize workload orchestration based on task processing latency and avoid the loss of tasks due to resource exhaustion. At time-step $t$, we define the system as a tuple $\langle W, \dot{W}, \dot{C}, \mathcal{C}, T_t \rangle$. Each node can decide to process a task locally or offload it to a neighbor, represented by the action variable $a_t^i$ for worker $i$. The delay incurred by the decisions of all agents is given by:

$$D_n(T_t, \dot{W}) = \sum_{a_t^n} d(a_t^n) \tag{2}$$

The function $d(a_t^n)$ represents the local extra delay of the decision made by agent $n$, defined as:

$$d(a_t^n) = \chi_D^{\text{wait}} T_{i,a_t^n}^{\text{wait}}(\tau_k) + \chi_D^{\text{comm}} T_{i,a_t^n}^{\text{comm}}(\alpha_k^{\text{out}}) + \chi_D^{\text{exc}} T_{i,a_t^n}^{\text{exc}}(\tau_k). \tag{3}$$

This function is a weighted sum of three time-related terms associated with offloading decisions, based in Baek et al. (2019); Kumari et al. (2022). The delay function incorporates hyperparameters $\chi_D^{\text{wait}}, \chi_D^{\text{comm}}$, and $\chi_D^{\text{exc}} \geq 0$. The delay terms for a given action $a_t^n$ are:

$$T_{\dot{w}^n, a_t^n}^{\text{wait}}(\tau_k) = \frac{Q_n^t}{N_\phi^n \phi_n} + \sum_{j \neq n} \frac{Q_j}{N_\phi^j \phi_j} I_j(a_t^n), \tag{4}$$

which represents the waiting time for task $\tau_k$ in the queue of node $W_i$ (and $W_j$ in case it is offloaded). Here, $\phi_i$ is the computing service rate of node $W_i$, $Q_n^t$ is the queue size of the same at time $t$, and $N_\phi^i$ is its number of processors. The indicator function $I_j(a_t^n)$ equals 1 if the task is processed locally on node $w_n$(i.e., $I_n(a_t^n) = 1$) or offloaded to a neighboring node $W_j$ (i.e., $I_j(a_t^n) = 1$) with $j \neq n$. The term $T_{i,a_t}^{\mathrm{comm}}(\alpha_k^{\mathrm{out}})$ denotes the communication cost of TO, defined as a delay (eq. 1), where $a_t^n$ indicates the neighboring node $i$. If the task is executed locally, this term becomes zero. The term:

$$T_{i,a_t^n}^{\mathrm{exc}}(\tau_k) = \frac{t\rho_k\xi_k}{N_\phi^{a_t^n}\phi_{a_t^n}} - \frac{t\rho_k\xi_k}{N_\phi^n\phi_n} \tag{5}$$

represents the difference in execution costs for tasks processed locally versus those processed at the target node. Here, $\rho_k$ denotes the number of instructions per task, and $\xi_k$ represents the number of CPU cycles per instruction. Hence, to minimize the delay in processing the tasks at each time-step, we wish to find the solution to the constrained optimization problem:

$$\min_{\{a_t^n\}_{\dot{w}_n \in \dot{W}}} \quad D(T_t, \dot{W}) \tag{6}$$

$$\text{subject to} \quad C_1 : \delta_i \leq t_C \tag{7}$$

$$C_2 : Q^n \leq Q_{max}^n \tag{8}$$

The solution must also respect a set of constraints to minimize task drops: no tasks may be offloaded to overloaded nodes, as indicated by constraint eq. 7, and no tasks should breach their deadlines. Additionally, no node should exceed its computational resource limit, as outlined in eq. 8. Although the constraints are enforced by dynamics of the environment, in practice, we relax this constraints to penalties to the objective.

**Partially-Observable Markov Game**. To solve the TO problem with distributed and decentralized agents, we define it as a Partially-Observable Markov Game (POMG) Hu et al. (2024), represented as a tuple $\langle \mathcal{N}, \mathcal{S}, \mathcal{O}, \Omega, \mathcal{A}, P, R \rangle$. Here, $\mathcal{N} = 1, \ldots, n$ denotes a finite set of agents; $\mathcal{S}$ is the global state space that includes the information about all the nodes and tasks in the network; $\Omega = \{o_i\}_{i \in \mathcal{N}}$ is the set of Observation Spaces, where $o_i$ is the observation space of agent $i$, that has information about the workers in its neighborhood, $\dot{N}_n$; $\mathcal{O} = \{O_i\}_{i \in \mathcal{N}}$ s.t.$O_i : S \to o_i$ is the set of Observation Functions for each agent, where $O_i$ is the observation function of agent $i$. The observation function maps the state to the observations for each agent. Each agent's observations includes information about its local computational and communication resources, the information shared by its neighbors on the same, and information about the next offloadable task. The details on the observation space are provided in 10.2. $\mathcal{A} = \{A_i\}_{i \in \mathcal{N}}$ is the set of action spaces, where $A_i$ is the action space of agent $i$. Each agent is able to select whether to send a task to one of its neighbors or process it locally; $P : S \times \mathcal{A} \to S$ is the unknown global state transition function; Lastly, $R = \{R_i\}_{i \in \mathcal{N}}$ s.t.$R_i \in S \times \mathcal{A} \times S \to \mathbb{R}$ - is the reward function for agent $i$. Each agent will consider the local reward given by:

$$R_i(s_t, a_t^i) = U - d(a_t^i) + \chi_O O(s_t, a_t^i) \tag{9}$$

Where $U$ is a constant utility term, $\chi_O \geq 0$ is a weighting parameter, and the term $\chi_O O(s_t, a_t^i)$ is the distance to overload the workers involved in an offloading, we define $O(s_t, a_t) = -\log(p_t^{Oa_t})/3$. And, $p_t^{Oa_t} = max(0, \frac{Q_{a_t}^{\mathrm{max}} - Q_{a_t}}{Q_{a_t}^{\mathrm{max}}})$, represents the distance to overloading node $W_i$, and $Q'_{a_t} = min(max(0, Q_{a_t} - \phi_{a_t}) + 1, Q_{a_t}^{\mathrm{max}})$ is the expected state of the queue at node $W_{a_t}$, after taking action $a_t$.

## 3 FAUNO

We now present FAuNO—Federated Asynchronous Network Orchestration—a framework designed to provide remote computing power to a group of clients, while load balancing in a decentralized fashion with an FRL-based algorithm. The FAuNO nodes also act as workers. A detailed breakdown of FAuNO node components is provided in annex 8.

## 3.1 FEDERATED REINFORCEMENT LEARNING TASK OFFLOADING SOLUTION

We consider two components to our solution: a local component and a global component. The local component utilizes Proximal Policy Optimization (PPO) Schulman et al. (2017). This algorithm belongs to the Actor-Critic family of algorithms, meaning that there is an actor component that learns to interact directly with the environment and a critic component that learns to evaluate the actor and guides the training. We federated the critic network in our solution so that the experience of all the agents is used to guide the local learning of the agents. Our global solution for training the critic network builds on FedBuff Nguyen et al. (2021), a buffered asynchronous aggregation method that we adapt to RL by allowing agents to keep training and sending updates to the global critic without stopping after the first round. This prevents stragglers from blocking progress while still incorporating shared updates into the global critic. The crux of the proposed algorithm is that by federating the global network, we mitigate selfish behavior among agents and improve sample efficiency through continuous, non-blocking training.

**Local Policy Optimization**   Our local optimization uses an adaptation of PPO to FL, combined with Generalized Advantage Estimation (GAE) Schulman et al. (2018) for computing advantages. In our version, agents independently interact with the environment and, after a configurable number of training steps, share their latest critic network with the global manager. Upon receiving an updated global model, each agent incorporates it as the next critic for training. The local optimization procedure is summarized in algo. 1, with full details provided in annex 11.1.

**Global Algorithm**   The Global Algorithm is responsible for managing the federation and aligning the local solutions from each participant to derive the global solution. We employ a non-blocking semi-asynchronous method to tackle the following optimization problem:

$$\min_w f(w), \text{ s.t. } f(w) := \frac{1}{m} \sum_{k=1}^{m} p_k l_k(w; \theta_k). \tag{10}$$

Here, $m$ represents the number of participants, and $\theta_k$ are the parameters of the actor-network for the agent identified by $k$. The variable $w$ corresponds to the global critic parameters, and $p_k$ is the weight assigned to agent $k$'s loss function. In our algorithm, $l_k$ is equivalent to the symmetric of eq. 41.

FAuNO's semi-asynchronous design addresses heterogeneity and stragglers. Faster agents contribute updates more frequently, while slower ones do not block progress. The gradients are buffered at the global manager (GM), with newer updates from the same agent replacing older ones and increasing the weight of that agent's last update in the aggregation. The weights are computed following algo. 3. The global critic is updated once updates from $K$ distinct agents are received. This allows for agents to continue training without waiting for the global training round to complete, allowing for continuous training even under straggling devices. To mitigate policy divergence and allow for specialization on each node as well, we federate only the critic network, while keeping the actors local. Furthermore, each agent's observation space is standardized and includes its own queue size, neighbors' queue and capacity states, aggregate task instruction counts, and features of the next task to be processed, more details in annex 10.2. Upon aggregation, the GM updates the global critic via a weighted average:

$$\hat{w} = w + \sum_{k \in \bar{K}} p_k \nabla w_k, \tag{11}$$

where $\bar{K} \geq K$ is the set of buffered updates. The coefficient $p_k$ is calculated based on the number of update steps each agent performed, ensuring that $\sum_{k \in \bar{K}} p_k = 1$.

Fig. 2 provides an overview of the FAuNO training global model training process. The global algorithm can be observed in algo. 2 in annex 9.

## 4 PERFORMANCE EVALUATION

In this section, we evaluate FAuNO's performance using two standard TO metrics: average task completion time and percentage of completed tasks—the proportion of tasks that were created and whose results were successfully returned to the originating client. We compare FAuNO against

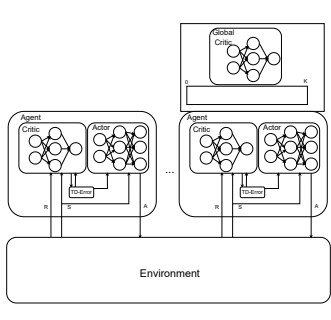

(a) Agents independently train local actor–critic models. Following algo 1

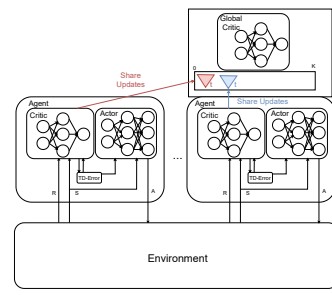

(b) Agents asynchronously transmit their critic updates to the Global Manager, which stores them in a buffer.

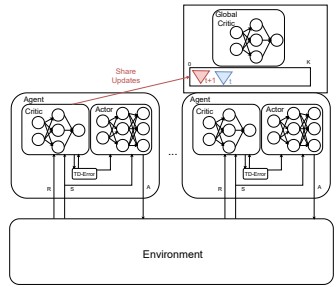

(c) When a new update from an already known agent arrives, the older entry in the buffer is replaced, and the agent's weight in the upcoming aggregation is increased according to its training steps.

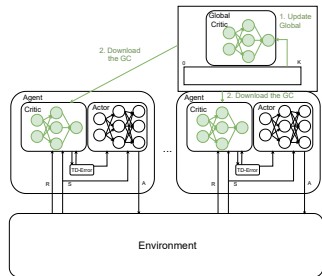

(d) Once updates from $K$ distinct agents are available, these are aggregated following eq. equation 11, producing a new global critic that is then redistributed to all agents.

Figure 2: The figure illustrates the sequence of operations carried out by the Global Manager (GM) during the asynchronous aggregation of local critic updates in FAuNO. Each agent trains its actor–critic model locally and periodically transmits only the critic weights to the GM. The GM buffers received updates, retaining only the most recent one from each agent and weighting it proportionally to the number of local steps performed. Once updates from at least $K$ distinct agents are collected, a FedAvg-style aggregation is applied to update the global critic, which is then redistributed to all agents for continued local training.

two baseline algorithms: Least Queues(LQ), which offloads tasks to the worker with the shortest queue, and an adaptation of the synchronous FRL solution, SCOF Peng et al. (2024). Since no public implementation of SCOF was available, we reimplemented the algorithm and released the code in FAuNO's repository. For fairness, we adapted SCOF to our observation and reward spaces and disabled its DP component, as privacy was not the focus of this work, and DP typically reduces accuracy. These adaptations ensure comparability without diminishing SCOF's core capabilities. Details on the baselines are provided in annex 11.2. We benchmark our solution using PeersimGym Metelo et al. (2024), with realistic topologies generated by the Ether tool Rausch et al. (2020). These are structured as hierarchical star topologies, where a stronger server provides resources to a small set of client nodes, and a more powerful central server supports the intermediate servers. We also evaluate on synthetic topologies composed of 10 and 15 nodes randomly distributed in a 100×100 area. In these settings, the number of high-capacity nodes remains fixed, while the number of client nodes increases. All tests use realistic task distributions enabled by PeersimGym's integration with the Alibaba Cluster Trace workload generator Tian et al. (2019), which we have rescaled to better suit the considered edge devices. Each algorithm is trained for 40 episodes, for a total of 400,000 steps, and evaluated during training all presented results are the average result for the metric in question across the 40 episodes. Finally, we present an ablation study on the impact of agent heterogeneity on convergence and the impact of reducing the $K$ parameter on the performance of the algorithm; due to space constraints, the ablation on the $K$ parameter is in annex 6. Additional details on the testing setup, topologies, workloads, and hyperparameters used are provided in annexes 11 and 12.

## 4.1 ETHER BASED TOPOLOGIES

Table 2: Finished Tasks Ratio

| Algorithm | $\lambda = 0.5$ | | $\lambda = 1$ | | $\lambda = 2$ | |
| --- | --- | --- | --- | --- | --- | --- |
| | 2 | 4 | 2 | 4 | 2 | 4 |
| FAuNO | **0.973±0.008** | **0.974±0.006** | **0.964±0.008** | **0.965±0.005** | **0.920±0.015** | **0.926±0.010** |
| SCOF | 0.943±0.030 | 0.940±0.022 | 0.860±0.052 | 0.832±0.035 | 0.715±0.057 | 0.666±0.045 |
| LeastQueues | 0.943±0.004 | 0.948±0.003 | 0.928±0.004 | 0.934±0.003 | 0.909±0.005 | 0.914±0.006 |

Table 3: Response Time (in simulation ticks)

| Algorithm | $\lambda = 0.5$ | | $\lambda = 1$ | | $\lambda = 2$ | |
| --- | --- | --- | --- | --- | --- | --- |
| | 2 | 4 | 2 | 4 | 2 | 4 |
| FAuNO | 170.208±13.573 | 165.174±7.775 | 204.282±14.168 | 199.448±9.045 | 262.350±13.103 | 246.271±8.745 |
| SCOF | 56.393±18.287 | 46.237±15.163 | 153.042±22.472 | 144.737±20.229 | 109.865±25.088 | 85.381±22.444 |
| LeastQueues | 258.412±9.073 | 239.590±6.635 | 278.219±8.926 | 256.989±7.658 | 296.598±7.790 | 269.761±5.071 |

Tab. 2 and 3 report the average percentage of completed tasks and the average task response time for each algorithm, across varying topologies and task arrival rates ($\lambda$). A general trend is that performance degrades as the number of nodes and $\lambda$ increase, primarily due to faster exhaustion of computational and shared resources (e.g., cloudlets). Despite this, FAuNO consistently achieves the highest task completion rates in most scenarios and outperforms the heuristic baselines in response time. Although SCOF achieves lower response times, it does so at the cost of significantly reduced task completion. We attribute this to the heterogeneity of the nodes and the reduced sample efficiency of the syncronous FRL, making it so that using a single global network without the local specialization sets an orchestration strategy that is too general, which leads to offloading from high-capacity nodes when they fill up, leading to task expiration and exclusion from the response time.

## 4.2 RANDOM TOPOLOGY

Table 4: Finished Tasks Ratio

| Algorithm | $\lambda = 0.5$ | | $\lambda = 1$ | | $\lambda = 2$ | |
| --- | --- | --- | --- | --- | --- | --- |
| | 10 | 15 | 10 | 15 | 10 | 15 |
| FAuNO | 0.882±0.049 | 0.770±0.042 | 0.764±0.068 | 0.557±0.038 | 0.570±0.050 | 0.335±0.021 |
| SCOF | 0.862±0.066 | 0.673±0.058 | 0.684±0.064 | 0.435±0.045 | 0.465±0.062 | 0.251±0.029 |
| LeastQueues | 0.946±0.008 | 0.855±0.022 | 0.928±0.009 | 0.665±0.030 | 0.803±0.052 | 0.358±0.026 |

Table 5: Response Time (in simulation ticks)

| Algorithm | $\lambda = 0.5$ | | $\lambda = 1$ | | $\lambda = 2$ | |
| --- | --- | --- | --- | --- | --- | --- |
| | 10 | 15 | 10 | 15 | 10 | 15 |
| FAuNO | **307.222±19.105** | 489.358±14.685 | 365.190±20.482 | 539.846±12.691 | 416.433±13.899 | 513.129±9.378 |
| SCOF | 318.740±19.275 | 428.266±22.082 | 348.579±18.172 | 416.573±27.193 | 365.207±16.676 | 388.341±18.723 |
| LeastQueues | 425.877±24.507 | 603.833±21.076 | 493.780±22.037 | 619.894±10.063 | 576.914±23.472 | 522.105±9.968 |

As in the Ether networks, increasing the network size significantly degrades performance in both task completion rate(tab. 4 and response time (tab. 5). This effect is exacerbated by the considered topology maintaining a fixed number of cloudlets while increasing the number of client nodes. In contrast to the more structured topology with a single cloudlet, the LQ algorithm outperforms FAuNO in task completion. This can be explained by the larger accessibility to more powerful nodes in the random topology, leading to more offloading and concurrent tasks being processed. A deeper analysis of the impact of topology is available in 6.3. However, LQ's disregard for local processing capabilities results in substantially higher response times. SCOF exhibits the opposite behavior: due to the presence of more powerful nodes distributed across the network compared to the Ether scenario, its centralized, non-personalized orchestration favors local processing. This reduces communication overhead and improves response time, but at the expense of lower task completion. FAuNO achieves a balanced trade-off between the two metrics. Moreover, the higher task completion rate of LQ in random topologies follows from the abundance of remote computational resources afforded by dense connectivity. Under high task arrival rates, local nodes saturate quickly and offloading becomes the only effective option. In this setting, LQ can exploit the large number of available remote queues,

Table 6: Global disagreement score (↓ better)

| Variant ↓ / Packet-drop D → | 0.3 | 0.5 | 0.8 |
|---|---|---|---|
| **FAuNO vs FAuNO** | 74.5 | 232.7 | 289.8 |
| **FAuNO vs Oracle critic** | 63.6 | 240.1 | 307.2 |
| **Pure MARL** ($D = 1$) | 147.54 | 262.22 | 313.81 |

Table 7: Disagreement scores. MARL (packet-drop rate 1.0) vs. centralized; FAuNO not shown

| Variant | Global disagreement $\delta$ |
|---|---|
| **Pure MARL vs Pure MARL** | 319.35 |
| **Fully centralized oracle vs MARL** | 325.55 |

preventing any single destination from becoming heavily congested. As a result, this experiment serves as a strong stress test, effectively benchmarking FAuNO against a heuristic whose behavior aligns closely with the structure of the random topology.

## 4.3 LEARNING UNDER HETEROGENITY

To evaluate FAuNO's stability under asynchronous, non-IID conditions, we designed a heterogeneous workload experiment using the 15-node random topology. The network was partitioned into three regions, each configured to process a specific workload class with different task sizes and arrival rates. Clients in each region generated tasks only from their corresponding class distribution; the details on the experiment, algorithms considered and the result analysis are available in the annex 6.1. We compared three training setups: FAuNO, pure MARL PPO, where agents learn without any shared critic, and a centralized oracle where all nodes share a single critic model. To assess consistency between the different critics, we introduced a critic-agreement protocol and measured critic consistency using a global disagreement score (eq. 13), $\delta$, based on pairwise RMSE across sampled states (eq. 12). During each evaluation episode, we sampled 500 global states and collected observations from each agent. To correct for the fact that agents processing faster task streams naturally accumulate higher rewards, all values were normalized by the corresponding task arrival rate before computing disagreement. Results are summarized in Tab. 6 and 7. As expected, disagreement between critics decreases when the packet-drop rate is reduced, indicating more consistent models as communication becomes more reliable. FAuNO's critics approach the predictions of the centralized oracle at low drop rates, confirming that aggregation yields stable shared learning. By contrast, the pure MARL variant showed substantially higher disagreement, highlighting its divergence under heterogeneous workloads. These results confirm that FAuNO is robust to non-IID conditions and mitigates policy inconsistency even when agents face systematically different task distributions.

## 5 CONCLUSION, LIMITATIONS & FUTURE WORK

We addressed the decentralized TO problem in edge systems by modeling it as a cooperative objective over a federation of agents, formalized within a POMG. To this end, we proposed **FAuNO**, a novel FRL framework that integrates buffered semi-asynchronous aggregation with local PPO-based training. FAuNO enables decentralized agents to learn task assignment and resource usage strategies under partial observability and limited communication, while maintaining global coordination through a federated critic. Empirical evaluation in the PeersimGym environment confirms FAuNO's superiority over heuristic and FRL baselines in terms of task loss and latency, highlighting its adaptability to dynamic and heterogeneous edge settings.

**Limitations.** From a security perspective, the current formulation assumes that all nodes are honest and cooperative. Adversarial and Byzantine behavior, although likely to occur in real-world edge environments, is not considered, and privacy preservation is also outside the present scope. At the system level, we consider only simulated environments, and we model a stable network with reliable nodes and communication links, excluding failures, congestion, and bandwidth constraints. Furthermore, we do not consider energy costs that could trade off with latency. These assumptions simplify the evaluation but omit factors critical to practical edge deployments. Algorithmically, the leveraging of FedBuff introduces a bias toward faster clients, potentially underrepresenting slower nodes. Moreover, reliance on a GM creates a single point of failure and a potential bottleneck in very large networks. Lastly, this study relies exclusively on simulation and does not include deployment on real edge infrastructures.

**Future Work.** Future work will focus on relaxing current assumptions to better match practical edge deployments, enabling a transition toward using FAuNO in real edge environments. This includes

supporting dynamic topologies and node mobility, handling intermittent connectivity, and addressing adversarial participation. At the system level, we plan to extend our objective to consider data locality, fault tolerance, and energy-consumption. Algorithmically, we will address the single-manager bottleneck by exploring hierarchical or decentralized critics to improve scalability and robustness. We also intend to incorporate energy-aware objectives to capture trade-offs between latency and resource use, and to design defenses against malicious agents to enhance security. Lastly, we plan to investigate the theoretical properties of semi-asynchronous federated actor–critic learning, including the effects of staleness, non-IID workloads, and delayed aggregation on convergence and value-function bias.

## REPRODUCIBILITY STATEMENT

We provide an anonymous repository at https://anonymous.4open.science/r/FAuNO-C976, which contains the complete codebase developed for this paper. This includes implementations of all proposed methods, test configurations, hyperparameters, and supporting scripts required to re-run the experiments and reproduce the reported results. Furthermore, the detailed implementation choices, hyperparameters, and training configurations are also partially documented in annex 11. The annex further includes descriptions of the experimental setup and evaluation protocol. Together, these materials are intended to enable full reproducibility of our results.

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

# 6 ABLATIONS

## 6.1 HETEROGENEOUS SETTING EXPERIMENT

We designed a heterogeneous workload experiment to evaluate whether FAuNO's federated critic converges stably under asynchronous, non-IID conditions, the precise setting where policy inconsistency would arise. We set specific regions of the network to handle different task types and have different cadences of arriving applications, $\lambda$.

We focus on the 15-node random topology detailed in 12, partitioned into 3 regions. Each of these regions is designed to process a specific workload class with different task types varying in number of instructions, $\rho$ in MBytes, data size, $\alpha^{\text{in}}$ and task arrival rate $\lambda$. We define each of the workload classes as:

- **W1** ($\lambda$=0.5; $\rho$=38,$\alpha^{\text{in}}$=32e7) this class is composed of nuc:2, rpi5 8G:2, rpi5 6G:1
- **W2** ($\lambda$=1; $\rho$=16; $\alpha^{\text{in}}$=64e7) this class is composed of nuc:2, rpi5 8G:2, rpi4:1
- **W3** ($\lambda$=2.0; $\rho$=64; $\alpha^{\text{in}}$=16e7) this class is composed of nuc:1, rpi5 6G:2, rpi4:2

A client attached to node $i$ draws tasks only from that node's class distribution.

**Training variants to compare**

- FAuNO (federated critic)
- Pure MARL PPO, agents train only on local observations with no shared global model (implemented by discarding all critic updates so that no global model learning occurs)
- Centralized oracle, where all participants share a single global critic network that is trained directly by all the participants without a decentralized global learning.

**Critic-agreement protocol**

1. **Evaluation set:** For $M$ states, $\{s_m\}$, collect the observations, $o_i(s_m)$, of each agent of that given state. During an evaluation episode, we sample around $M = 500$ distinct global states.

2. **Value Prediction matrix:** We then build matrix $V \in \mathbb{R}^{N \times M}$, where line $i$ represents the evaluation of critic $i$ and column $m$ is the evaluation of point $m$. Thus, entry $V_{i,m}$ is defined as

$$V_{i,m} = V_i(o_i(s_m))$$

And, we also built the $V_\star$ matrix with the evaluation of the centralized critic model for the same observations.

3. **Pair-wise RMSE:** We then compute our metrics. A matrix where for each agent, we have the Root Mean Squared Error (RMSE) between the different state evaluations. This metric is meant to capture the differences on evaluating the observations by each of the agents. Each entry of this matrix is given by:

$$\text{RMSE}_{ij} = \sqrt{\frac{1}{M} \sum_m \left(V_{i,m} - V_{j,m}\right)^2}. \tag{12}$$

Where the $V_{i,m}$ is the evaluation of agent i, to provide a global metric of divergence of the matrix, we consider the global disagreement score computed as:

$$\delta = \frac{2}{N(N-1)} \sum_{i<j} \text{RMSE}_{ij}. \tag{13}$$

These same metrics are then re-computed for $V_\star$.

**Comparison of the value function between nodes and against the** $V_*$    The group with a higher rate of task arrival $\lambda$ will have better chances of increasing the received reward; thus, the estimate of the value will not be comparable. To have a fair measure of the quality between the different agent groups, we use the following normalization, for every node $i$:

$$\tilde{V}_{i,m} = V_{i,m}/\lambda_i$$

We study the global disagreement score, $\delta$, across packet drop rates $D$. As expected, disagreement decreases as $D$ is reduced (tab.6), reflecting more consistent agents. The federated critic increasingly aligns across the network as communication becomes more reliable. tab.6 shows that this alignment approaches the centralized oracle's predictions at low $D$, confirming that FAuNO benefits from improved communication and maintains stable shared learning even in highly heterogeneous settings. In contrast, tab.7 demonstrates that the MARL variant suffers from substantially higher disagreement under the same conditions. We must disclose that, although the MARL agent had begun converging, training was not completed but we expect the divergence to increase with training.

This analysis confirms that FAuNO's federated critic is robust to non-IID workloads and can mitigate policy inconsistency even when agents face systematically different distributions.

## 6.2   K Exploration

Table 8: Straggler and aggregation-threshold ablation results (mean ± s.d.)

| Setting | Finished-task ratio | Avg. response time (ticks) |
|---|---|---|
| K = 0.3, s = 0.3 | $0.77 \pm 0.02$ | $258.85 \pm 5.95$ |
| K = 0.3, s = 0.5 | $0.76 \pm 0.02$ | $259.74 \pm 6.91$ |
| K = 0.3, s = 0.8 | $0.76 \pm 0.02$ | $260.07 \pm 7.02$ |
| K = 0.5, s = 0.3 | $0.76 \pm 0.02$ | $259.09 \pm 6.35$ |
| K = 0.5, s = 0.5 | $0.76 \pm 0.02$ | $258.83 \pm 7.71$ |
| K = 0.5, s = 0.8 | $0.76 \pm 0.02$ | $260.86 \pm 6.97$ |

We evaluated FAuNO's robustness under straggler conditions and varying aggregation thresholds. Let $s$ denote the fraction of gradients dropped before reaching the global node. In the straggler test, the objective is not to improve performance metrics but to avoid collapse. FAuNO achieves this goal: even when 80% of gradients are dropped ($s = 0.8$), both the finished-task ratio and average response time remain well within one standard deviation of their baseline values. This demonstrates that FAuNO gracefully degrades to local MARL when global connectivity is severely reduced, maintaining essentially the same performance. We also examined buffer sensitivity by doubling the aggregation threshold from $K = 0.3$ to $0.5$. This change affects performance by less than 1% in either metric, indicating that FAuNO is robust to variations in buffer size under this workload.

## 6.3   Impact of topology on performance on synthetic networks

Table 9: Connection counts by topology and node type

| Connection Type | Random (10 nodes) | Random (15 nodes) | 2 Clusters (23 nodes) | 4 Clusters (45 nodes) |
|---|---|---|---|---|
| nuc–nuc | 2.4 | 2.4 | 0 | 0 |
| nuc–rpi | 3.2 | 8.4 | 8 | 8 |
| rpi–nuc | 3.2 | 4.2 | 1 | 1 |
| rpi–rpi | 3.2 | 7.6 | 0 | 0 |
| srv–nuc | 0 | 0 | 2 | 4 |
| srv–rpi | 0 | 0 | 16 | 32 |

The stronger task completion rate of LQ in random topologies reflects important topology-specific dynamics. As shown in Table 9, the random topologies considered in our experiments exhibit high connectivity between weaker RPIs and stronger NUCs (e.g., approximately 8.4 connections per RPI in the random topology with 15 nodes). This connectivity enables offloading with relatively low delay and cost, helping avoid congestion at any single NUC, which in turn makes LQ essentially optimal in

these networks. With abundant remote computation capacity and limited contention, benchmarking against LQ in such settings effectively pushes FAuNO to its limits.

In contrast, our reward shaping was designed to discourage excessive offloading to mitigate latency and congestion in the star-like topologies generated by Ether. This global reward structure thus introduces a bias toward local processing. LQ, being reactive and unconstrained by this reward design, offloads more aggressively and achieves higher task throughput, albeit with higher delay. This illustrates a throughput–latency trade-off shaped jointly by topology and reward structure.

## 6.4 ABLATION ON THE FEDERATED COMPONENTS

In this section, we compare several configurations of the global optimization layer. Alongside the FAuNO setup presented in the main text, we evaluate three additional variants: a synchronous version of FAuNO, a configuration that federates only the actor, and a configuration that federates both actor and critic. All variants follow the same overall optimization procedure, differing only in the parameters participating in global aggregation.

In the actor-only variant, the global objective is:

$$\min_{\theta} \ f(\theta), \quad f(\theta) = \frac{1}{m} \sum_{k=1}^{m} p_k \, l_k(\theta; \, w_k), \tag{14}$$

and aggregation proceeds analogously to Eq. equation 11, but applied to the actor parameters:

$$\hat{\theta} = \theta + \sum_{\theta \in \bar{K}} p_k \, \nabla w_k. \tag{15}$$

In the actor–critic variant, both parameter sets participate in the global update:

$$\min_{w, \theta} \ f(w, \theta), \quad f(w, \theta) = \frac{1}{m} \sum_{k=1}^{m} p_k \, l_k(w, \theta), \tag{16}$$

with actor and critic updates aggregated using equation 15 and equation 11.

In the synchronous variant, agents only perform local training while holding the most recent global model, and aggregation is triggered solely after a full round of updates has been received.

These variants allow us to directly assess the individual contributions of critic-only federation and the semi-asynchronous buffer mechanism by contrasting FAuNO with actor-only, actor–critic, and fully synchronous alternatives.

### 6.4.1 EXPERIMENTS

**Random Topology Results**

These experiments use the weight configuration from Table 31, rely on the alternative problem formulation in Appendix 2.2, and follow the same experimental setup described in Section 4.2.

Tables 10 and 11 present the finished-task ratio and response-time results for the different aggregation strategies in the random-topology setting. The results show that FAuNO achieves a higher task completion rate than the variant that federates only the actor. While the two variants exhibit comparable response-time behavior, the higher completion ratio indicates that FAuNO makes more effective use of the available computational resources. For this reason, we consider FAuNO the preferable configuration among the evaluated options.

Table 10: Finished Tasks Ratio

| Algorithm | $\lambda = 0.5$ | | $\lambda = 1$ | | $\lambda = 2$ | |
|---|---|---|---|---|---|---|
| | 2 | 4 | 2 | 4 | 2 | 4 |
| FAuNO | 0.882±0.049 | **0.770±0.042** | **0.764±0.068** | **0.557±0.038** | **0.570±0.050** | **0.335±0.021** |
| FAuNO (Fed. Actor) | 0.888±0.044 | 0.756±0.054 | 0.741±0.068 | 0.505±0.034 | 0.542±0.053 | 0.306±0.018 |
| FAuNO (Fed. Actor & Critic) | 0.890±0.057 | 0.759±0.049 | 0.734±0.075 | 0.518±0.045 | 0.560±0.050 | 0.318±0.022 |
| FAuNO (Sync) | 0.877±0.055 | 0.763±0.042 | 0.746±0.055 | 0.530±0.042 | 0.560±0.053 | 0.324±0.023 |

Table 11: Response Time (in simulation ticks)

| Algorithm | $\lambda = 0.5$ | | $\lambda = 1$ | | $\lambda = 2$ | |
| --- | --- | --- | --- | --- | --- | --- |
| | 10 | 15 | 10 | 15 | 10 | 15 |
| FAuNO | **307.222±19.105** | 489.358±14.685 | 365.190±20.482 | 539.846±12.691 | 416.433±13.899 | 513.129±9.378 |
| FAuNO (Fed. Actor) | 308.576±16.320 | 469.830±11.346 | 362.134±12.131 | 496.677±10.633 | 405.929±12.692 | 474.001±10.559 |
| FAuNO (Fed. Actor & Critic) | 314.765±18.595 | 474.993±11.876 | 367.260±13.635 | 491.064±11.397 | 404.677±10.772 | 477.402±7.83 |
| FAuNO (Sync) | 302.760±19.873 | 474.553±13.187 | 363.699±13.945 | 512.065±10.156 | 412.764±13.968 | 486.376±11.241 |

## Ether Topology Results

These experiments use the weight configuration from Table 31, rely on the alternative problem formulation in Appendix 2.2, and follow the same experimental setup described in Section 4.2. In the Ether-based topologies, FAuNO clearly outperforms the federated-actor variant, and matches or outperforms both the actor-critic or the synchronous variant, achieving substantially higher throughput when the number of tasks arriving becomes higher and the risk of overloading increases. We attribute the actor-only performance gap to the sensitivity of the actor network to weight perturbations introduced during aggregation, which appears to hinder stable policy improvement when the actor is federated.

Table 12: Finished Tasks Ratio

| Algorithm | $\lambda = 0.5$ | | $\lambda = 1$ | | $\lambda = 2$ | |
| --- | --- | --- | --- | --- | --- | --- |
| | 2 | 4 | 2 | 4 | 2 | 4 |
| FAuNO | 0.973±0.008 | 0.974±0.006 | **0.964±0.008** | **0.965±0.005** | **0.920±0.015** | **0.926±0.010** |
| FAuNO (Fed. Actor) | 0.959±0.021 | 0.966±0.015 | 0.891±0.038 | 0.896±0.020 | 0.754±0.049 | 0.736±0.027 |
| FAuNO (Actor & Critic) | 0.971±0.012 | 0.974±0.006 | 0.932±0.020 | 0.940±0.012 | 0.833±0.035 | 0.848±0.026 |
| FAuNO (Sync) | **0.975±0.008** | **0.976±0.007** | 0.943±0.017 | 0.950±0.012 | 0.868±0.024 | 0.875±0.013 |

Table 13: Response Time (in simulation ticks)

| Algorithm | $\lambda = 0.5$ | | $\lambda = 1$ | | $\lambda = 2$ | |
| --- | --- | --- | --- | --- | --- | --- |
| | 2 | 4 | 2 | 4 | 2 | 4 |
| FAuNO | 170.208±13.573 | 165.174±7.775 | 204.282±14.168 | 199.448±9.045 | 262.350±13.103 | 246.271±8.745 |
| FAuNO (Actor) | 75.027±10.810 | 62.660±11.372 | 109.899±11.867 | 97.184±6.892 | 143.984±10.680 | 123.519±5.377 |
| FAuNO (Actor & Critic) | 112.681±12.550 | 105.716±10.889 | 148.427±15.277 | 138.236±10.152 | 198.605±15.209 | 192.819±11.451 |
| FAuNO (Sync) | 117.291±13.444 | 102.204±11.706 | 160.493±14.854 | 141.316±13.852 | 222.994±14.357 | 198.281±10.543 |

## 6.5 COMPARING ALIGNMENT ALGORITHMS

In this section we benchmark FAuNO's aggregation mechanism. We incorporated into FAuNO a FedProx Li et al. (2020) style proximal term to the objective solved locally by each of the agents:

$$\Psi(w, w_0) = \frac{\mu}{2} \|w_k - w\|^2, \qquad (17)$$

between the local critic weights, $w_k$, and the global critic weights, $w$, during the computing of the agent gradients when computing the local agent objectives. Originating the follwing objective,

$$L_t^F(\theta, w) = \mathbb{E}_t \left[ L_t^{\text{CLIP}}(\theta) - c_1 L_t^{\text{VF}}(w) + c_2 S_{\pi_\theta}(s_t) + \Psi(w, w_0) \right]. \qquad (18)$$

Where the $w_0$ is a copy of the latest pulled global model. We will henceforth refer to standardt FAuNO as simply FAuNO, and the FAuNO trained with the proximal term as FAuNOProx. For all experiments we considered a $\mu = 0.005$.

### 6.5.1 EXPERIMENTS

Tables 14 and 15 report the comparison between the FAuNO and the FAuNOProx in the random topologies, and tables 16 and 17 report the comparison between FAuNO and FAuNOProx in the ether-based topolgies. These experiments follow the same experimental setup described in Section 4.1 and Section 4.2 respectively. In the random-topology experiments, FAuNOProx underperforms relative to FAuNO consistently processing less tasks, although we still observe the trade-off between throughput and response time. A plausible explanation for this results, is the structure of the observation space (Section 10.2), which is designed to provide structurally similar state representations across agents. In this setting, the proximal term in FedProx, that focuses in handling non-i.i.d state constructions, offers limited benefit and may restrict useful updates updates collected from different nodes.

**Ether topology results:**

Table 14: Finished Tasks Ratio

| Algorithm | $\lambda = 0.5$ | | $\lambda = 1$ | | $\lambda = 2$ | |
|---|---|---|---|---|---|---|
| | 2 | 4 | 2 | 4 | 2 | 4 |
| FAuNO | **0.973±0.008** | **0.974±0.006** | **0.964±0.008** | **0.965±0.005** | **0.920±0.015** | **0.926±0.010** |
| FAuNO(PROX) | 0.964±0.016 | 0.967±0.009 | 0.923±0.023 | 0.918±0.021 | 0.807±0.047 | 0.823±0.030 |

Table 15: Response Time (in simulation ticks)

| Algorithm | $\lambda = 0.5$ | | $\lambda = 1$ | | $\lambda = 2$ | |
|---|---|---|---|---|---|---|
| | 2 | 4 | 2 | 4 | 2 | 4 |
| FAuNO | **170.208±13.573** | **165.174±7.775** | **204.282±14.168** | 199.448±9.045 | 262.350±13.103 | 246.271±8.745 |
| FAuNO(PROX) | 180.040±13.590 | 174.239±8.066 | 212.129±8.242 | **197.196±7.443** | **246.862±7.644** | **231.045±5.480** |

**Random topology results:**

Table 16: Finished Tasks Ratio

| Algorithm | $\lambda = 0.5$ | | $\lambda = 1$ | | $\lambda = 2$ | |
|---|---|---|---|---|---|---|
| | 10 | 15 | 10 | 15 | 10 | 15 |
| FAuNO | **0.882±0.049** | **0.770±0.042** | **0.764±0.068** | **0.557±0.038** | **0.570±0.050** | **0.335±0.021** |
| FAuNO(PROX) | 0.853±0.072 | 0.697±0.067 | 0.675±0.093 | 0.475±0.042 | 0.493±0.066 | 0.276±0.028 |

Table 17: Response Time (in simulation ticks)

| Algorithm | $\lambda = 0.5$ | | $\lambda = 1$ | | $\lambda = 2$ | |
|---|---|---|---|---|---|---|
| | 10 | 15 | 10 | 15 | 10 | 15 |
| FAuNO | **307.222±19.105** | 489.358±14.685 | 365.190±20.482 | 539.846±12.691 | 416.433±13.899 | 513.129±9.378 |
| FAuNO(PROX) | 328.024±12.126 | **475.331±18.708** | **360.794±15.652** | **484.038±14.355** | **382.343±19.206** | **442.796±11.775** |

## 6.6 ALTERNATIVE REWARD FORMULATION

To complement the shaped-reward experiments, we also include a sparse reward baseline that retains only task-level utility and drop penalties. This minimal formulation provides a lower-information setting that helps assess FAuNO's stability when the reward signal is significantly reduced.

### 6.6.1 PROBLEM FORMULATION

We aim to optimize workload orchestration based on task processing latency and avoid the loss of tasks due to resource exhaustion. At time-step $t$, we define the system as a tuple $\langle W, \dot{W}, \dot{C}, C, T_t \rangle$. Each node can decide to process a task locally or offload it to a neighbor, represented by the action variable $a_t^i$ for worker $i$. To encourage efficient task orchestration, we define an objective function that focuses on maximizing serviced clients. At each time-step $t$, the reward is

$$R_t = U_t - D_t, \tag{19}$$

where $U_t = \alpha \bar{\tau}_c$ is the utility from completed tasks, $D_t = \alpha \bar{\tau}_d$ is the penalty from dropped tasks, and $F_t$ is a reward shaping component. Here, $\bar{\tau}_c$ and $\bar{\tau}_d$ denote the number of tasks completed and dropped since the previous step, and $\alpha$ is a task-level reward unit.

The objective at each step is therefore to select node-level actions that maximize $R_t$, balancing throughput, latency, and stability of the system. This induces the constrained optimization problem:

$$\max_{\{a_t^n\}_{\dot{w}_n \in \dot{W}}} R_t \tag{20}$$

$$\text{subject to} \quad C_1: \ \delta_i \leq t_C \tag{21}$$

$$C_2: \ Q^n \leq Q_{\max}^n \tag{22}$$

where $C_1$ prohibits offloading toward nodes violating their operational limits (preventing propagation of overload), and $C_2$ constrains queue sizes to mitigate excessive service delays. These constraints are enforced by the environment but translated into penalties within the reward to ensure smooth optimization during training.

**Partially-Observable Markov Game**. To solve the TO problem with distributed and decentralized agents, we define it as a Partially-Observable Markov Game (POMG) Hu et al. (2024), represented as a tuple $\langle \mathcal{N}, \mathcal{S}, \mathcal{O}, \Omega, \mathcal{A}, P, R \rangle$. Here, $\mathcal{N} = 1, \ldots, n$ denotes a finite set of agents; $\mathcal{S}$ is the global state space that includes the information about all the nodes and tasks in the network; $\Omega = \{o_i\}_{i \in \mathcal{N}}$ is the set of Observation Spaces, where $o_i$ is the observation space of agent $i$, that has information about the workers in its neighborhood, $\dot{N}_n$; $\mathcal{O} = \{O_i\}_{i \in \mathcal{N}}$ s.t. $O_i : S \to o_i$ is the set of Observation Functions for each agent, where $O_i$ is the observation function of agent $i$. The observation function maps the state to the observations for each agent. Each agent's observations includes information about its local computational and communication resources, the information shared by its neighbors on the same, and information about the next offloadable task. The details on the observation space are provided in 10.2. $\mathcal{A} = \{A_i\}_{i \in \mathcal{N}}$ is the set of action spaces, where $A_i$ is the action space of agent $i$. Each agent is able to select whether to send a task to one of its neighbors or process it locally; $P : S \times \mathcal{A} \to S$ is the unknown global state transition function; Lastly, $R = \{R_i\}_{i \in \mathcal{N}}$ s.t. $R_i \in \mathcal{O} \times \mathcal{A} \times \mathcal{O} \to \mathbb{R}$ - is the reward function for agent $i$. Each agent will consider the local reward given by:

$$R_i(o_i^t, a_t^i, o_i^{t+1}) = U_t^i - D_t^i + F_t^i(o_i^t, o_i^{t-1}), \tag{23}$$

where a potential-based reward shaping term is added to the base objective in 9. The shaping component is

$$F_t^i(o_i^t, o_i^{t-1}) = \Phi_t^i - \Phi_{t-1}^i, \tag{24}$$

where the potential at time-step $t$ for node $i$ is defined as

$$\Phi_t^i = w_g \left( T_r^{i,t} + T_p^{i,t} + T_c^{i,t} + T_o^{i,t} \right), \tag{25}$$

with $w_g$ a global weighting factor. Each term captures a distinct operational property of the node:

$$T_r^{i,t} = \frac{w_r}{1 + \bar{R}_t^i}, \tag{26}$$

which penalizes increases in the node $i$'s average response time $\bar{R}^{i,t}$, defined as the mean completion delay of tasks processed by the node.

$$T_p^{i,t} = w_f \frac{\mathcal{T}_p^{i,t}}{1 + \mathcal{T}_d^{i,t} + \mathcal{T}_p^{i,t}}, \tag{27}$$

which promotes higher throughput by increasing the relative proportion of completed tasks. Here $\mathcal{T}_p^{i,t}$ is the total number of processed tasks at the node and $\mathcal{T}_d^{i,t}$ the total number of dropped tasks by the node.

$$T_c^{i,t} = w_c \, \mathcal{T}_O^{i,t}, \tag{28}$$

which penalizes overloads, where $\mathcal{T}_O^{i,t}$ is the number of overload occurrences recorded at the node.

$$T_o^{i,t} = -w_o \frac{Q_t^i}{Q_{\max}^i}, \tag{29}$$

a direct penalty on queue occupancy, where $Q_t^i$ is the queue length and $Q_{\max}^i$ the maximum queue capacity of node $i$. The coefficients $w_r$, $w_o$, $w_f$, and $w_c$ control the relative importance of the response-time, queue-load, throughput, and overload contributions within the potential function.

### 6.6.2 EXPERIMENT RESULTS

Tables 19 and 18 present the results for the ether-based topologies, and Tables 20 and 21 report the results for the random topologies, other than the objective being solved we follow the test setup from section 4. Using the reduced-information reward, FAuNO obtains lower task-completion ratios than LQ, even in the ether setting, but continues to outperform SCOF. This behaviour is consistent with the minimal structure of the sparse reward even with the reward shaping, which provides limited feedback on queue dynamics, overload conditions, or local resource constraints. Both heuristic methods exploit their fixed decision rules independently of the reward, whereas FAuNO receives substantially less guidance for learning effective offloading strategies.

In the ether topologies, the simplified reward leads to faster processing due to the inherent through-put–latency trade-off, but the lack of shaping prevents FAuNO from matching the heuristics' ability to capitalize on available remote resources. In the random topologies, the heuristic methods continue to benefit from dense connectivity, while FAuNO maintains stable training under the sparse objective.

Overall, these experiments serve as a sparse-reward baseline demonstrating FAuNO's behaviour under minimal feedback and highlighting the role of reward structure in guiding learning under partial observability.

**Results for the ether based topologies**

Table 18: Finished Tasks (in simulation ticks)

| Algorithm | $\lambda = 0.5$ | | $\lambda = 1$ | | $\lambda = 2$ | |
|---|---|---|---|---|---|---|
| | 2 | 4 | 2 | 4 | 2 | 4 |
| FAuNO | **0.971±0.012** | **0.970±0.008** | 0.921±0.024 | 0.920±0.019 | 0.809±0.037 | 0.810±0.031 |
| SCOF | 0.937±0.027 | 0.931±0.024 | 0.868±0.050 | 0.832±0.048 | 0.720±0.050 | 0.668±0.049 |
| LeastQueues | 0.943±0.004 | 0.948±0.003 | 0.928±0.004 | 0.934±0.003 | 0.909±0.005 | 0.914±0.006 |

Table 19: Response Time (in simulation ticks)

| Algorithm | $\lambda = 0.5$ | | $\lambda = 1$ | | $\lambda = 2$ | |
|---|---|---|---|---|---|---|
| | 2 | 4 | 2 | 4 | 2 | 4 |
| FAuNO | **116.350±24.117** | **128.639±18.481** | 176.122±13.143 | 183.136±11.347 | 234.280±7.513 | 228.712±5.787 |
| SCOF | 141.254±38.572 | 135.879±24.767 | 77.623±23.393 | 60.140±14.816 | 132.559±21.705 | 87.377±21.738 |
| LeastQueues | 258.412±9.073 | 239.590±6.635 | 278.219±8.926 | 256.989±7.658 | 296.598±7.790 | 269.761±5.071 |

**Results for the random topologies**

Table 20: Finished Tasks (in simulation ticks)

| Algorithm | $\lambda = 0.5$ | | $\lambda = 1$ | | $\lambda = 2$ | |
|---|---|---|---|---|---|---|
| | 10 | 15 | 10 | 15 | 10 | 15 |
| FAuNO | 0.837±0.082 | 0.694±0.069 | 0.676±0.089 | 0.467±0.038 | 0.496±0.061 | 0.285±0.025 |
| SCOF | 0.858±0.069 | 0.690±0.054 | 0.675±0.064 | 0.428±0.047 | 0.482±0.062 | 0.243±0.034 |
| LeastQueues | 0.946±0.008 | 0.855±0.022 | 0.928±0.009 | 0.665±0.030 | 0.803±0.052 | 0.358±0.026 |

Table 21: Response Time (in simulation ticks)

| Algorithm | $\lambda = 0.5$ | | $\lambda = 1$ | | $\lambda = 2$ | |
|---|---|---|---|---|---|---|
| | 10 | 15 | 10 | 15 | 10 | 15 |
| FAuNO | 313.700±14.388 | 478.446±18.919 | 356.469±17.983 | 486.970±16.111 | 388.021±16.474 | 445.638±11.494 |
| SCOF | 306.651±22.589 | 413.587±20.072 | 344.903±16.542 | 411.175±31.781 | 373.415±17.686 | 395.833±14.413 |
| LeastQueues | 425.877±24.507 | 603.833±21.076 | 493.780±22.037 | 619.894±10.063 | 576.914±23.472 | 522.105±9.968 |

## 6.7 ABLATION AGAINS MULTI-AGENT ALGORITHMS WITH GLOBAL STATES

In this ablation, we compare FAuNO with a centralized MARL baseline, MAPPO Yu et al. (2022), under conditions that favor the latter. MAPPO serves as an oracle benchmark in which each agent has access to the full global state rather than its local observation. For a fully equitable comparison, the environment would need to propagate global state information through the transition dynamics; however, for the purpose of this study we evaluate MAPPO using complete state information without having to send the state information through the network to establish an upper bound on performance.

Our implementation follows the variant of MAPPO in which each agent maintains its own actor and critic networks. The critic receives the full global state, enabling it to condition on the observations of all agents, while the actor conditions only on the agent's local information. This contrasts with FAuNO, where critics are federated using local observations and actors remain fully decentralized. The MAPPO configuration therefore provides an oracle reference point illustrating the performance obtainable when global information is directly available to the value function, we note that in a real system this information would need to be aggregated at the centralized node, before any decision could be made.

### 6.7.1 GLOBAL OBSERVATION MODELLING

For the oracle experiments, we construct a global state that aggregates all node-level information normally accessible only partially accessible to each node through their local observations. At time step $t$, the global state contains, for every node $W^p$, its queue size at the time-step $Q_t^p$, its identifier $n^p$, and information about the next task scheduled for processing, $\tau_t^i$. Formally, the observation for each agent is constructed by:

$$O_{i,t}^g = \langle \tau_t^i\, n^1,\, Q_t^1,\, n^2,\, Q_t^2,\, \ldots,\, n^{|W|},\, Q_t^{|W|},\, \rangle. \tag{30}$$

Where $\tau_t^p$ encodes the attributes of the next task at node $i$:

$$\tau_t^p = (\mathbb{I}_{\text{local}}^p,\, \rho_t^p,\, \alpha_p^{\text{instr}},\, \alpha_p^{\text{in}},\, \alpha_p^{\text{out}}). \tag{31}$$

Here, $\mathbb{I}_{\text{local}}^p$ indicates whether the next task is assigned for local execution, $\rho_t^p$ is the task's current progress, $\alpha_p^{\text{instr}}$ is the total required instructions, and $\alpha_p^{\text{in}}$ and $\alpha_p^{\text{out}}$ are its input and output sizes.

In contrast to the local observation space, the global state does not require padding or placeholder values, since it provides complete system information independent of network connectivity. This representation is used for the oracle MAPPO, which conditions on the full system state rather than localized observations.

### 6.7.2 EXPERIMENT RESULTS

Tables 23 and 22 present the results for the ether-based topologies, and Tables 24 and 25 report the results for the random topologies. Other than the objective being solved, we follow the same experimental setup as in Section 4.

As expected, the model with access to global observations outperforms FAuNO. Since MAPPO conditions its critic on the full system state, it can be regarded as an upper bound on the performance achievable by any PPO-based decentralized method in this environment. In this light, the gap between MAPPO and FAuNO provides a direct measure of the value of global information.

Across the ether topologies, the performance of FAuNO remains close to that of the oracle MAPPO baseline. Task-completion rates differ only marginally in low and moderate load regimes, and response times remain within a similar range despite FAuNO operating solely from local observations. This suggests that the federated critic is able to approximate the global value signal effectively enough to guide decentralized actors toward near-oracle behavior. In higher-load settings, MAPPO maintains a clearer advantage, which is consistent with the regime where global state information yields stronger gains. Even so, FAuNO continues to track the oracle closely rather than collapsing under partial observability.

Overall, FAuNO's results lie near those of the full-information baseline in the ether topologies, indicating that the semi-asynchronous federated critic can recover much of the benefit of global coordination without requiring system-wide visibility.

**Results for the ether based topologies**

Table 22: Finished Tasks (in simulation ticks)

| Algorithm | $\lambda = 0.5$ | | $\lambda = 1$ | | $\lambda = 2$ | |
|---|---|---|---|---|---|---|
| | 2 | 4 | 2 | 4 | 2 | 4 |
| FAuNO | 0.971±0.012 | 0.970±0.008 | 0.921±0.024 | 0.920±0.019 | 0.809±0.037 | 0.810±0.031 |
| MAPPO | 0.976±0.012 | 0.977±0.006 | 0.955±0.013 | 0.952±0.009 | 0.878±0.022 | 0.885±0.015 |

Table 23: Response Time (in simulation ticks)

| Algorithm | $\lambda = 0.5$ | | $\lambda = 1$ | | $\lambda = 2$ | |
|---|---|---|---|---|---|---|
| | 2 | 4 | 2 | 4 | 2 | 4 |
| FAuNO | 116.350±24.117 | 128.639±18.481 | 176.122±13.143 | 183.136±11.347 | 234.280±7.513 | 228.712±5.787 |
| MAPPO | 114.903±22.825 | 98.887±8.966 | 143.937±16.653 | 137.958±11.061 | 219.118±15.619 | 203.494±10.227 |

**Results for the random topologies**

Table 24: Finished Tasks (in simulation ticks)

| Algorithm | $\lambda = 0.5$ | | $\lambda = 1$ | | $\lambda = 2$ | |
| --- | --- | --- | --- | --- | --- | --- |
| | 10 | 15 | 10 | 15 | 10 | 15 |
| FAuNO | 0.837±0.082 | 0.694±0.069 | 0.676±0.089 | 0.467±0.038 | 0.496±0.061 | 0.285±0.025 |
| MAPPO | N/A | N/A | N/A | N/A | N/A | N/A |

Table 25: Response Time (in simulation ticks)

| Algorithm | $\lambda = 0.5$ | | $\lambda = 1$ | | $\lambda = 2$ | |
| --- | --- | --- | --- | --- | --- | --- |
| | 10 | 15 | 10 | 15 | 10 | 15 |
| FAuNO | 313.700±14.388 | 478.446±18.919 | 356.469±17.983 | 486.970±16.111 | 388.021±16.474 | 445.638±11.494 |
| MAPPO | N/A | N/A | N/A | N/A | N/A | N/A |

## 7 EXTENDING PEERSIMGYM

The PeersimGym Metelo et al. (2024) environment for TO with multi-agent reinforcement learning was not originally designed for federated learning. To address this, we extended it with the FL Updates Manager (FLManager) to enable the exchange of FL updates across the simulated network. The FL process begins with the FL algorithm determining which updates to share. These updates are sent to the environment, where the FLManager generates an ID for each update, calculates its size, and stores the relevant information. This data is then transmitted to the simulation, which sends a dummy message with the size of the update from the node hosting the source agent to the node hosting the destination agent through the network. FL agents can then query the FLManager for completed updates, prompting it to retrieve any updates that have traversed the simulated network. To ensure compatibility with other environments, we decoupled the FLManager from PeersimGym and introduced a customizable mechanism for computing the number of steps an update takes to arrive. The code for the FLManager is available in the FAuNO repository (https://anonymous.4open.science/r/FAuNO-C976; anonymized).

## 8 FAuNO NODES

Each FAuNO node consists of three key components: the orchestration agent or manager, the information exchange module, and the resource provisioning component. As our focus is on developing an algorithm for the decentralized orchestration of clients' computational requirements, we keep the other components generic for adaptability across various scenarios. As illustrated in Fig. 3, one of the participants assumes the role of FAuNO GM, managing the global model; this role can be assumed by any node in the network. In our experiments, the data processing and collection layers are built into the simulation.

## 9 ALGORITHMS

We provide the pseudocode for the two phases of FAuNO learning. We also provide the actual code developed in our git repository (https://anonymous.4open.science/r/FAuNO-C976; anonymized). The first component we mention is the Local algorithm, as seen in the algo. 1 ran by the participants in the Federation:

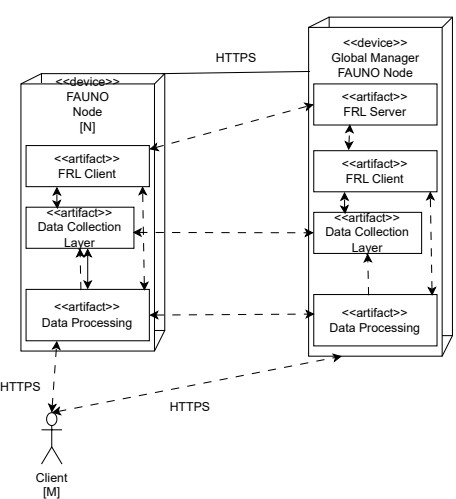

Figure 3: Deployment diagram representing the different components in the basic and GM FAuNO Nodes, with the arrow representing that some messages are exchanged between the nodes.

---

**Algorithm 1** FAuNOLocalPPO

---

**Require:** Initial critic weights $w_0$, learning rate local critic $\eta_{\text{critic}}$, learning rate local actor $\eta_{\text{actor}}$, initial actor weights $\theta_0$, minibatch size $M$, number of steps between trainings $N$, number of steps before sharing weights with global $T$

1: $\theta_{\text{old}} \leftarrow \theta_0$
2: $w \leftarrow w_0$
3: steps $\leftarrow 0$
4: version $\leftarrow 0$
5: **for** iteration $= 1, 2, \ldots$ **do**
6:    $w$, version $\leftarrow$ `checkIfNewerGlobalArrived()` {Resets number of steps since last update}
7:    **for** step $= 1, 2, \ldots, N$ **do**
8:       Run policy $\pi_{\theta_{\text{old}}}$ in environment for timesteps
9:    **end for**
10:   Compute advantage estimates $\hat{A}_1, \ldots, \hat{A}_T$ using $V(\cdot; w)$
11:   Optimize surrogate $L_t^F$ w.r.t. $\theta$ and $w$, with $K$ epochs and minibatch size $M$
12:   $\theta_{\text{old}} \leftarrow \theta$
13:   steps $\leftarrow$ steps $+ 1$
14:   **if** iteration $\mod T = 0$ **then**
15:      `shareUpdatesWithGlobal(`$\nabla w$, steps, version`)` {Asynchronous operation}
16:   **end if**
17: **end for**

---

In this algorithm, the *checkIfNewerGlobalArrived()* function checks whether a newer version of the global critic model has been sent. If so, it returns the updated model; otherwise, it returns the current model, $w$, that the agent has trained. Similarly, the *shareUpdatesWithGlobal(u, steps, version)* function asynchronously shares the latest updates, $u$, with the global node. We note that minimizing the negative of eq. 41, $-L_t^F$, is equivalent to maximizing the original objective.

Then we look at the global algorithm executed by one of the nodes in the federation in algo. 2.

---

**Algorithm 2** FAuNOGlobalManager

---

**Require:** Global critic learning rate $\eta_{\text{critic}}$, local actor learning rate $\eta_{\text{actor}}$, client training steps $Q$, buffer size $K$, all participating agents $m$, minibatch size $M$, number of steps between trainings $N$, number of steps before sharing weights with global $T$
**Ensure:** FL-trained global critic model $w_g$
1: $w_g \leftarrow w_0$
2: Initialize Buffer $\leftarrow \{\}$ {Start with an empty buffer}
3: $k \leftarrow 0$
4: **while** not converged **do**
5:     Run FAuNOLocalPPO($w_0, \eta_{\text{critic}}, \eta_{\text{actor}}, \theta_0, M, N, T$) on $m$ {Asynchronous operation}
6:     **if** client update received and used latest $k$ **then**
7:         Receive $\Delta_i$, steps$_i$, version$_i$ from client $i$
8:         **if** $\Delta_i \notin$ Buffer **then**
9:           Add $\Delta_i$, steps$_i$, version$_i$ to Buffer
10:          $k \leftarrow k + 1$
11:        **else if** steps$_i$ > steps stored in Buffer **then**
12:          Replace $\Delta_i$ in Buffer with the newer one
13:        **end if**
14:        **if** $k \geq K$ **then**
15:          $w_g \leftarrow w_g + \sum_{k \in \text{Buffer}} \text{computeCoefficient}(\text{Buffer}, k)\Delta_k$
16:          Clear Buffer
17:          $k \leftarrow 0$
18:          sendLatestModelToClients() {Asynchronous operation}
19:        **end if**
20:     **end if**
21: **end while**

---

In this algorithm, *computeCoefficient()* calculates the weight of each update based on the number of updates each agent sent, see 3, and *sendLatestModelToClients()* is a method that sends the latest global critic network to all the clients.

---

**Algorithm 3** computeCoefficient

---

**Require:** Buffer with updates Buffer, target agent $k$
**Ensure:** Coefficient of agent $k$'s update
1: $total\_k \leftarrow 0$
2: no_steps $\leftarrow 0$
3: **for** each entry $i \in$ Buffer **do**
4:     agent$_i \leftarrow$ agent that sent entry $i$
5:     steps$_i \leftarrow$ steps performed in $i$'s update
6:     $total\_k \leftarrow total\_k +$ steps$_i$
7:     **if** $k ==$ agent$_i$ **then**
8:         no_steps $\leftarrow$ steps$_i$
9:     **end if**
10: **end for**
11: **return** no_steps$/total\_k$

---

## 10 PEERSIMGYM ENVIRONMENT AND THE POMG

### 10.1 COMMUNICATION PROTOCOLS

There are three types of messages shared between the nodes. These are

- **Exchange of information to neighbors** - Our framework assumes that each node can share its local state only with directly connected neighbors through low-overhead broadcast or multicast mechanisms. This realistically mirrors real-world edge deployments, where full global state observability is impractical due to network size, reliability, and cost constraints.

- **Exchange of tasks** - Our framework assumes that tasks can be offloaded between directly connected nodes within their neighborhood, allowing localized workload distribution without reliance on centralized coordination.

- **Exchange of federated updates** - Model updates are propagated to the global node through multihop communication when direct connectivity is not available.

## 10.2 Observation Space

The observation space for agent $p$ in node $W^p$ at time step $t$ consists of it's own queue size $Q_t^p$, the latest queue size known for each of it's neighbors $\{Q_t^j | W_j \in \tilde{W}^p\}$, where $\tilde{W}_p$ is the node $W^p$'s neighborhood, and the percentage of space free for itself, $F_t^p$, and each of the neighbors, $F_t^j$, computed as:

$$F^n = Q_t^n / Q_{max}^n \tag{32}$$

Then on the task dimension they observe information about the tasks to be processed in particular the total number of instructions in the queue given by eq. 33, the total number of instructions of tasks assigned to be processed locally given by eq. 34 where $\mathbb{I}_{\text{local}_p}(\tau^i)$ is the identifier whether task $\tau^i$ was assigned to be processed locally in node $p$. Lastly, we have information on the next task to be processed, namely, its id $i$, the current progress of the task at time-step $t$, $\rho_t^i$, the total instructions, and the data input, $\alpha_i^{in}$, and output size, $\alpha_i^{out}$.

$$Q_{\rho,t}^p = \sum_{\tau_i \in Q_p} \rho^i \tag{33}$$

$$Q_{\rho,t}^{p,\text{local}} = \sum_{\tau_i \in Q_p} \mathbb{I}_{\text{local}}(\tau^i) \tag{34}$$

$$\mathbb{I}_{\text{local}_p}(\tau^i) = \begin{cases} \rho^i, & \text{if } \tau^i \text{ is local} \\ 0, & \text{otherwise} \end{cases} \tag{35}$$

Moreover, we convert the observations of all the agents to be structurally similar by normalizing and padding the observation spaces, ensuring consistent input dimensionality and robustness to network topological changes or node failures. Specifically, missing neighbors are represented using normalized placeholder values (-1), maintaining stable critic evaluation despite node heterogeneity.

## 10.3 Simulaiton Realism

To mitigate the reality gap inherent to simulation-based evaluation, we leverage PeersimGymMetelo et al. (2024) together with the two integrated tools, grounded in real-world data. First, workloads are generated using traces derived from Alibaba CloudTian et al. (2019), providing CPU, memory, and temporal profiles that reflect actual cluster behavior. Second, topologies are synthesized with Ether Rausch et al. (2020), which produces edge-computing infrastructures aligned with real edge deployment scenarios, such as urban sensing. Together, these components supply PeersimGym with realistic load patterns, resource variability, and communication structures, enabling evaluation conditions that approximate an operational edge system.

## 11 Implementation Details

### 11.1 PPO Formulation

PPO Schulman et al. (2017) is a policy gradient method grounded in the Policy Gradient Theorem Sutton et al. (1999), which enables training a policy approximator by estimating the policy gradient and applying stochastic gradient ascent:

$$\hat{g} = \mathbb{E}_t \left[ \nabla_\theta \log \pi_\theta(a_t \mid s_t) \hat{A}_t^{GAE} \right] \tag{36}$$

Here, $\pi_\theta$ is the policy being optimized, and $\hat{A}_t$ is an estimator for the Advantage Function computed with Generalized Advantage Estimation Schulman et al. (2018).

$$\hat{A}_t^{GAE} = \sum_{l=0}^{k-1} (\lambda\gamma)^l \delta_{t+l}^V \tag{37}$$

$$s.t.\ \delta_t^V = -V(s_t) + \sum_{i=0}^{k-1} \gamma^k r_{t+i}, \tag{38}$$

Here, $k$ can vary from state to state and is upper-bounded by a parametrized value, $N$, while $V(\cdot)$ would be an estimator for the value function.

The PPO algorithms work by running a policy for a parametrizable number of steps and storing information not only about the state, action, and reward, but also about the probability assigned to the chosen action. This information is then utilized in the next training step for computing the objective function, which in the case of PPO-Clip, is given by:

$$L^{\text{CLIP}}(\theta) = \mathbb{E}_t\Big[ \min\Big( r_t(\theta)\hat{A}_t, \text{clip}(r_t(\theta), 1-\epsilon, 1+\epsilon)\hat{A}_t \Big)\Big] \tag{39}$$

Here, $r_t(\theta)$ denotes the probability ratio:

$$r_t(\theta) = \frac{\pi_\theta(a_t \mid s_t)}{\pi_{\theta_{\text{old}}}(a_t \mid s_t)}. \tag{40}$$

The rationale behind using the probability ratio is that when an action with a higher advantage is selected and the new policy assigns a higher probability to that action, then the ratio will be bigger than zero, obtaining an overall higher objective. Conversely, if the probability increases for a negative advantage, then the objective function decreases faster. The clipping and the minimum are set in place so that the final objective is a lower bound (i.e., a pessimistic bound) on the unclipped objective Schulman et al. (2017). This prevents excessive deviations from the original policy in a single update, avoiding large, harmful updates caused by outliers.

Due to the Actor-Critic nature of the PPO algorithm, two components must be trained: the critic and the actor. Consequently, when utilizing automatic differentiation frameworks, like PyTorch, Schulman et al. Schulman et al. (2017) recommend maximizing the following objective:

$$L_t^F(\theta, w) = \mathbb{E}_t\Big[ L_t^{\text{CLIP}}(\theta) - c_1 L_t^{\text{VF}}(w) + c_2 S_{\pi_\theta}(s_t) \Big], \tag{41}$$

where we have the objective of the Actor, $L^{\text{CLIP}}$, as shown in eq. 39. The critic's loss function, $L^{\text{VF}}(w)$, where $w$ is the parameters of the critic network, given by,

$$L_t^{\text{VF}}(w) = (r + \gamma V(s_{t+1}) - V(s_t))^2. \tag{42}$$

And an entropy term, $S[\pi_\theta](s_t)$ to promote exploration. The $c_1$ and $c_2$ are coefficients weighing the different components of the objective.

And, because we are exploring an FL approach, the agents will share the gradients they obtained while training the local critic networks following algo. 1 in annex 9.

## 11.2 BASELINES

To compare FAuNO, we implement a set of baseline policies. We classify these baseline policies into two different categories, the first is the heuristic baselines **Least Queue**, which selects the observable worker with the smallest queue size relative to its maximum queue size and offloads the next eligible task to that worker. The purpose of the heuristic baseline is to provide a reference point that is widely understood and accessible, serving as a benchmark for expected performance, offering a familiar comparison point that helps contextualize the results.

We then consider the State-of-the-art synchronous FRL algorithm, SCOF Peng et al. (2024), in the spirit of looking at the benefits of considering the improvements of an asynchronous training mechanism that keeps training even, considering heterogeneous devices and communication delays.

SCOF is an algorithm designed for TO in the IIoT setting with a focus on vertical offloading from Edge devices to a set of Edge Nodes from the SBCs, not considering the offloading mechanics of the Edge Nodes themselves. The algorithm itself considers a Federated Duelling DQN, that is aggregated with a FedAvg-based approach and utilizes differential privacy (DP) to improve the security of the update exchanges. We could not find any implementation of SCOF, so we provided our implementation of the algorithm based on SCOF's paper Peng et al. (2024) in FAuNO's repository. Since no public implementation of SCOF was available, we reimplemented the algorithm and released the code in FAuNO's repository. To ensure a fair comparison with FAuNO, which does not use DP, we disabled SCOF's DP component, as DP often reduces accuracy and was not the focus of this study. We further adapted SCOF to our Markov game formulation and edge setting, modifying the observation and reward structures for compatibility. These adaptations were applied consistently and do not disadvantage SCOF beyond removing features absent in FAuNO.

### 11.3 VERSIONING AND CONTROLLING STALE MODELS

Each global model carries a version identifier that tracks the most recent global update, this identifier is shared with the agents when they send the models to the global. Agents store this version locally and attach it to every update they send. The Global Manager accepts only updates matching the current global version, ensuring that stale updates derived from outdated models are discarded and freshness is maintained throughout training.

### 11.4 HYPERPARAMETERS USED FOR FAuNO

We based our choice of hyperparameters on Andrychowicz et al. (2020). The parameters used for FAuNO:

Table 26: Hyperparameters Used in FAuNO Experiments

| Parameter | Value | Explanation |
|---|---|---|
| $\gamma$ | 0.90 | Discount factor for the long-term reward computation |
| $\epsilon$ | 0.5 | PPO clipping parameter |
| $\eta$ | 0.00001 | Learning rate for the global model (affects critic) |
| $\mu$ | 0.005 | Scales the the proximal term in PPO |
| Actor Learning rate | 0.001 | Learning rate for the actor network |
| Critic Learning rate | 0.0003 | Learning rate for the critic network |
| Critic Loss coefficient | 0.5 | Coefficient for the critic loss term |
| Entropy Loss coefficient | 0.5 | Coefficient for the entropy loss term |
| Save interval | 1500 steps | Frequency at which models are saved |
| Steps per exchange | 150 steps | Number of steps before exchanging data |
| Steps per episode | 150 steps | Number of steps per training episode |
| Batch size | 30 | Size of batches for gradient updates |

For SCOF, we adopted the hyperparameter settings reported in the original paper Peng et al. (2024). For parameters not specified, we selected values empirically. A complete list of settings is provided in the repository under `configs/algo_configs`.

### 11.5 NETWORK ARCHITECTURE

The architectures for the PPO are based on the ones in Barhate (2024)

## 12 TEST SETUP

Here, we give the concrete simulation setup configurations and elaborate on the baseline algorithms used. More details are available in the repository FAuNO repository[2]

---

[2]https://anonymous.4open.science/r/FAuNO-C976/README.md

**Actor Network Architecture**    **Critic Network Architecture**

(a) Actor Network    (b) Critic Network

Figure 4: Actor-Critic Neural Network Representations

## 12.1 ETHER EXPERIMENT PARAMETERS

The experiments are based on two distinct network topologies generated using Ether, with 2 and 4 AoT clusters. Each simulated AoT cluster consists primarily of SBCs, modeled as Raspberry Pi 3s, along with a base station equipped with an Intel NUC and two GPU units, and a remote, more powerful server. The topologies vary in cluster numbers, ranging from one to four clusters, and correspondingly in node counts, from 12 to 31. The number of agents making task-offloading decisions scales with the number of nodes, with all SBCs, NUCs, and the remote server hosting an agent. This results in 10 to 23 agents across different topologies. We configure the simulation so that only the nodes at the edge of the network, the SBCs, will directly receive tasks. The specific number of each node type is available in tab. 27, and the number of nodes taking up a given function is available in tab. 28.

| No. Clusters | SBCs | NUCs | GPU units | Servers |
|:---:|:---:|:---:|:---:|:---:|
| 2 | 16 | 2 | 4 | 1 |
| 4 | 32 | 4 | 8 | 1 |

Table 27: Cluster Composition Table

| No. Clusters | No Agents | Nodes getting tasks from clients | Total nodes |
|:---:|:---:|:---:|:---:|
| 2 | 19 | 16 | 23 |
| 4 | 37 | 32 | 40 |

Table 28: Cluster Configuration Table

The visualization produced for each of the scenarios can be observed in fig.5a

Regarding the capabilities of the different components involved in the simulation, we relied on the hardware specifications generated by the Ether tool. We supplemented this information with data we found for each machine. This information is available in tab. 29.

Task generation at each SBC node follows a $Poisson(\lambda)$ distribution over a simulation episode of 1000 time steps, with each time step scaled by a factor of 10, making each tick equivalent to 1/10th of a second, for a total of 10,000 ticks per episode. Each agent makes an offloading decision at every time step, performing 30 episodes, with the ability to take action at each tick. A full list of the parameters used in our simulation can be found in tab. 30. We note that all time-dependent functions are scaled as well.

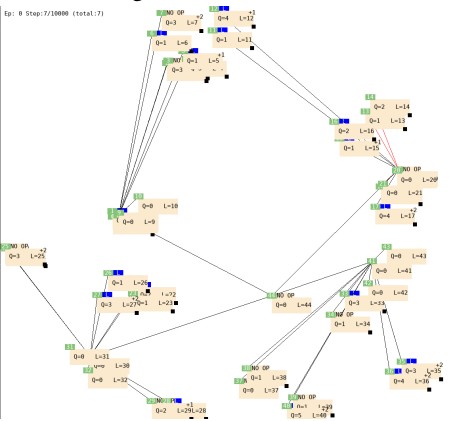

(a) Topology with 2 AoT clusters, consisting of 12 SBC, 2 NUCs, 4 GPU units, and 1 server. The total number of nodes is 19, and 15 agents manage task-offloading decisions, as outlined in tab. 27.

(b) Topology with 4 AoT clusters, consisting of 18 SBC, 4 NUCs, 8 GPU units, and 1 server. The total number of nodes is 31, with 23 agents managing task-offloading decisions, as shown in tab. 27.

Figure 5: Visualization of the different simulations used.

Table 29: Device Capacities

| Device | CPU (Millis) | Memory (Bytes) |
|---|---|---|
| Raspberry Pi 4 | 7200 | 6442450944 |
| Raspberry Pi 5 6GB | 9600 | 6442450944 |
| Raspberry Pi 6 8GB | 9600 | 8589934592 |
| Intel NUC | 14800 | 68719476736 |
| Cloudlet | 290400 | 188000000000 |

The tables 31 and 32 includes the reward weights used for the random topology experiments based on the eq.9 objective.

## 12.2 ARTIFICIAL NETWORK EXPERIMENT PARAMETERS

We consider two topologies with 10 and 15 nodes randomly distributed across a 100x100 square. These topologies have an increasing number of SBCs and a fixed number of NUCs. All SBCs receive tasks, and all the nodes in the topology have agents controlling them. The SBCs are picked randomly in equal proportions from the options in tab.29. The concrete number of nodes for each is given

Table 30: Parameter values in the experimental setup.

| Simulation time, $T$ | 1000 s | | |
|---|---|---|---|
| Task instructions, $\rho_i$ | $8 \times 10^7$ | Task utility, $\alpha$ | 100 |
| CPI, $\xi_i$ | 1 | Weight global, $w_g$ | 1 |
| Deadline, $\delta_i$ | 100 | Weight response time, $w_r$ | 1 |
| Bandwidth, $B_{i,j}$ | 4 MHz | Weight overloads, $w_c$ | 1 |
| Transmission power, $P_i$ | 40 dbm | Weight occupancy, $w_o$ | 1 |
| Scale | 10 | Weight processing, $w_p$ | 1 |

Table 31: Delay Weight Parameters

| Weight Parameter | Value |
|---|---|
| $\chi_D^{wait}$ | 0.6 |
| $\chi_D^{exc}$ | 1 |
| $\chi_D^{comm}$ | 1 |
| $\chi_O$ | 60 |

Table 32: Delay Weight Parameters

| Weight Parameter | Value |
|---|---|
| $\chi_D^{wait}$ | 0.6 |
| $\chi_D^{exc}$ | 1 |
| $\chi_D^{comm}$ | 0.4 |
| $\chi_O$ | 60 |

in tab. 33: We consider the same hyperparameters explained in 26. And consider similar training conditions to the ether-based topologies.

## 12.3 ALIBABA CLUSTER TRACE-BASED WORKLOAD

The workload considered for the experiments in the paper was based on the integration of Peer-simGym Metelo et al. (2024) with an Alibaba Cluster trace-based workload generation tool Tian et al. (2019). However, the original task sizes were unsuitable for the edge environment under study, particularly for client nodes, which became overwhelmed and dropped nearly 90% of tasks. To address this, we implemented a rescaling mechanism that adjusted the number of instructions per task while keeping all other characteristics the same. After evaluating several scaling factors, we selected a 10% reduction, which maintained a meaningful level of computational demand without causing excessive task loss.

## 12.4 COMPUTATIONAL REQUIREMENTS

The tests were all executed in a private High-Performance Computer, orchestrated by Slurm. Each Slurm job utilized 16 GB of RAM memory, 4 cores, and a MiG partition with one compute partition and 10 GB of memory of an Nvidia A100 GPU. Each of the tests that utilized a GPU took about 10 to 18 hours, depending on the number of agents, to complete the 400 000 steps.

## 13 USE OF LARGE LANGUAGE MODELS (LLMS)

Large language models were used solely as assistive tools to improve the clarity and readability of the manuscript. Their role was limited to editing for grammar, style, and wording. All research ideas, methodology, analysis, results, and conclusions were conceived and written by the authors. The authors carefully reviewed and verified all text to ensure accuracy and that the original meaning of the content was not violated, and we take full responsibility for the final content.

| No. Nodes | SBCs | NUCs |
|-----------|------|------|
| 10        | 5    | 5    |
| 15        | 10   | 5    |

Table 33: Cluster Composition Table

