# OpenReview forum: "FAuNO: Semi-Asynchronous Federated Reinforcement Learning Framework for Task Offloading in Edge Systems"
_ICLR.cc/2026/Conference — Submitted to ICLR 2026_

### Official Review · Reviewer_N43s · 2025-10-26

**Soundness:** 2
**Presentation:** 3
**Contribution:** 2
**Rating:** 4
**Confidence:** 4

**Summary:**

This paper presents a semi-asynchronous FRL framework tailored for decentralized Task Offloading (TO) in dynamic edge computing systems. The framework models the TO problem as a Partially Observable Markov Game (POMG).

**Strengths:**

The paper presents a novel adaptation of buffered semi-asynchronous FRL for edge task offloading. This design effectively mitigates straggler effects and enhances sample efficiency, which is critical in heterogeneous edge environments.
The federated critic / local actor architecture is well-constructed, balancing global coordination with local autonomy. It respects partial observability and improves robustness against node heterogeneity.

**Weaknesses:**

1. Single Point of Failure: The framework relies on a centralized global critic, which could become a bottleneck or a single point of failure in real-world deployments. A decentralized or hierarchical critic design might improve robustness.

2. Lack of Real-World Experiments: All evaluations are conducted in simulation (PeersimGym), which may not capture real-world complexities. Real-world experiments or deployment on testbeds would significantly strengthen the paper.

**Questions:**

How would FAuNO perform in scenarios with adversarial or unreliable nodes? Is there any mechanism to detect or mitigate poisoned updates?

---

> ### Author Response · Authors · 2025-12-03
>
> We thank the reviewer for the constructive and thoughtful feedback.
>
> We used the reviewer’s comments to reinforce the presentation of FAuNO’s architectural choices, clarify the scope of the method, and better justify the design decisions behind the federated critic. All revisions are highlighted in green in the updated manuscript.
>
> ## W1: Single Point of Failure
>
> > The framework relies on a centralized global critic, which may become a bottleneck or single point of failure; decentralized or hierarchical variants could improve robustness.
>
> We agree that a centralized global critic can become a bottleneck or single point of failure. This limitation is already noted in the manuscript (Limitations), and we now emphasize it more clearly by revising line ~482 (originally 478) to: “Moreover, relying on a global manager introduces a single point of failure and may limit scalability in very large networks.”
>
> In FAuNO, however, the goal is not to propose a failure-resilient FRL architecture. The primary technical contribution is a semi-asynchronous adaptation of FedBuff to reinforcement learning, with critic-only federation under partial observability. This design enables continuous local training, stable aggregation under heterogeneous device speeds, and decentralized execution via local actors.
>
> Future work (revised lines ~485) now explicitly motivates hierarchical or decentralized critics as a natural extension beyond the scope of this submission.
>
>
>
> ## W2: Lack of Real-World Experiments
> We agree that real-world deployments are ideal but fall outside the scope of this paper. We clarified two aspects in the revised paper:
> 1. Realism of the simulation environment.
> Section 10.3 now explains more clearly how PeersimGym[1] integrates:
>     - Alibaba Cluster Trace–based workloads[2], which provide task arrival patterns and resource profiles derived from production systems.
>     - Ether-generated topologies[3], which synthesize heterogeneous edge networks modeled on real infrastructures.
> 1. Scope and future directions.
> In the Limitations and Future Work sections, we explicitly state that physical testbeds are not included in this submission. We clarified that we are already developing the necessary infrastructure to move FAuNO toward real deployment.
>
> These points make explicit that while real-world experiments are not present, the simulation stack is grounded in realistic job traces and network structures, reducing the gap between simulation and deployment.
>
>
>
> ## Q1: Adversarial or Unreliable Nodes
>
> > How does FAuNO perform against adversarial nodes and poisioned updates?
>
> FAuNO is designed for cooperative, non-adversarial settings. The method does not include Byzantine-resilient aggregation or poisoned-update detection, and we clarify this directly in the revised Limitations section.
>
> That said, FAuNO incorporates a staleness-handling mechanism:
> - The global manager increments a version number after each aggregation;
>
> - Only updates carrying the latest version are accepted into the buffer;
>
> - Agents locally enforce the same mechanism to avoid overwriting with stale copies.
>
> This guards against inconsistency from unreliable or slow nodes, but not adversarial tampering. Addressing adversarial settings would require robust FL aggregation, anomaly detection, or trust scores, which we now cite as explicit future work rather than implicit assumptions.
>
> ___
>
> We appreciate the reviewer’s feedback. The revised paper addresses all raised concerns and significantly strengthens clarity, methodological soundness, and empirical rigor. We hope the improvements make the contributions of FAuNO clearer and demonstrate its relevance to FRL under realistic, heterogeneous edge-computing conditions.
>
>
> [1] Frederico Metelo, Cláudia Soares, Stevo Rackovic, and Pedro Ákos Costa. Peersimgym: An environment for solving the task offloading problem with reinforcement learning. In Machine Learning and Knowledge Discovery in Databases. Applied Data Science Track, pp. 38–54, Cham, 2024. Springer Nature Switzerland. ISBN 978-3-031-70378-2.
>
> [2] Huangshi Tian, Yunchuan Zheng, and Wei Wang. Characterizing and synthesizing task dependencies of data- parallel jobs in alibaba cloud. In Proc. ACM Symp. Cloud Comput., 2019.
>
> [3] Thomas Rausch, Clemens Lachner, Pantelis A Frangoudis, Philipp Raith, and Schahram Dustdar. Synthesizing plausible infrastructure configurations for evaluating edge computing systems. In 3rd USENIX Workshop HotEdge 20, 2020.

---

### Official Review · Reviewer_wmpm · 2025-10-27

**Soundness:** 3
**Presentation:** 4
**Contribution:** 3
**Rating:** 4
**Confidence:** 4

**Summary:**

This paper proposes FAuNO, a semi-asynchronous Federated Reinforcement Learning (FRL) framework for task offloading in edge systems.
The main idea is to extend FedBuff-style buffered asynchronous aggregation into an actor–critic setup: local agents train PPO-based actors, while a Global Manager (GM) asynchronously aggregates critic updates from clients.
The goal is to deal with stragglers and heterogeneity in edge systems by allowing faster nodes to contribute more frequently without blocking others.
The authors evaluate FAuNO in the PeersimGym simulator on two types of topologies (Ether-like clustered and random networks).
They compare against a simple heuristic (Least Queue) and a reimplemented synchronous federated RL baseline (SCOF).
They also include ablations on heterogeneity, packet-drop rates, and buffer-size thresholds.
Overall, FAuNO shows improvements over baselines in most scenarios, though performance depends on network topology.

**Strengths:**

- The paper addresses an important practical problem (handling stragglers in federated edge RL) with a sensible approach combining buffered asynchronous aggregation and actor-critic MARL.
- The presentation is generally clear, with helpful visualizations (Fig. 2) and a logical flow from problem formulation to experiments.
- The experimental evaluation includes multiple network topologies (Ether-based and random), ablations on buffer size (Table 8) and packet drops, and a heterogeneity analysis (Section 6.1).
- The authors transparently acknowledge current limitations (lines 471-485, 756-760), including the single global manager bottleneck and topology-specific reward tuning.
- The PeersimGym extension (Section 7) to support federated update exchange is a useful engineering contribution.

**Weaknesses:**

While FAuNO is cleanly implemented and well-motivated, a few important weaknesses keep it from being fully convincing in its current form:
1. Mathematical inconsistencies and unclear formulations.
   Several of the paper's core equations need revision or clarification:
   * The communication delay formula (Eq. 1) mixes logarithms with dB values, which gives units of bits/Hz rather than seconds. This should be fixed using $T = \frac{\alpha}{B \log_2(1+\text{SNR}_{\text{linear}})}$. Using natural log gives wrong capacity units.
   * The reward function (Eq. 9) adds delay as a *positive* term, which reverses the expected reward logic.
   * The global objective (Eq. 10) minimizes over critic weights $w$ even though $l_k(w, \theta_k)$ depends on both local and global parameters. The text should explain the alternating optimization scheme more clearly.
   * The advantage estimation (Eq. 20) is labeled as GAE, but it's actually an n-step formulation without λ. This is minor but worth correcting.
2. Narrow set of baselines.
   The comparisons include only LQ and SCOF. These are decent but relatively weak. Adding comparisons with more modern federated or asynchronous RL baselines (e.g., FedAsync, FedProx, MAPPO) would give a clearer sense of where FAuNO stands.
3. Reward shaping limits generalizability (though authors acknowledge this).
    The authors transparently admit that their reward function is "tailored to mitigate congestion in star-like Ether topologies" and "biases FAuNO toward local processing" in random networks. While this honesty is appreciated, it reveals a practical limitation: FAuNO's performance is sensitive to topology-specific reward tuning. In random topologies, LQ outperforms FAuNO on task completion (Table 5) precisely because the reward discourages offloading even when the network structure supports it. This suggests FAuNO may require manual reward retuning for each deployment scenario, limiting its plug-and-play applicability. Testing alternative reward formulations or learning adaptive reward weights would strengthen the generalizability claims.
4. Constraint vs. RL mismatch.
   The paper begins by framing the task offloading problem as a constrained optimization (Eqs. 6–8) but never enforces these constraints. In practice, FAuNO turns them into soft penalties in the reward. This is fine, but it should be clearly stated that the hard-constraint formulation is relaxed.
5. Observation normalization.
   Using −1 to represent missing neighbors while normalizing other inputs to [0, 1] is questionable. It's not serious, but a masking or zero-padding strategy would be better i think.
6. Reproducibility problem – repository link is empty.
   The paper provides an anonymous code link (https://anonymous.4open.science/r/FAuNO-C976) but the repository is empty and contains no runnable scripts. While the appendix includes detailed hyperparameters, the lack of working code prevents independent verification of results.

**Questions:**

1. Concerning Eq. 9, was the delay term negated during implementation (making a longer delay lead to less reward)?

2. In Eq. 10, when minimizing over the critic weights w, do you hold the actor parameters $\theta_k$ fixed?

3. How does FAuNO's performance change as the buffer threshold $K$ increases beyond 0.5?

4. Could the global critic aggregation be made hierarchical or fully decentralized?

5. The clip parameter in PPO is set to $\varepsilon = 0.5$, which is substantially greater than typical values (0.1–0.2). Was this for stability purposes?

---

> ### Author Response · Authors · 2025-12-03
>
> We thank the reviewer for their thoughtful comments.
>
> Their feedback allowed us to clarify several equations and notational choices, strengthen the presentation of the empirical evaluation, and refine key implementation details.
>
> Below we address the raised weaknesses (W) and questions (Q). Modifications to the paper colored in green.
>
> ## Regarding W1, Q1 and Q2: Clarifications and Coherence of the equations
>
> To address the reviewers concerns we have:
> > W1.1: Clarification of Log in eq.1
>
> The actual implementation of Eq. 1 used $log_2$ we changed the eq.1 to reflect this.
> > W1.2: Positive delay term in reward (Eq. 9)
> > Q1: Was the Eq. 9 delay term also flipped in the implementation?
>
> The actual reward considered the delay as a negative penalty. We have corrected Eq.9 to reflect this. We clarify the changes in line 257.
> > W1.3: Clarification to nature of variables in Eq.10.
> > Q2: Are the actor weights fixed in Eq.10?
>
> Yes, for the purposes of global aggregation the actor parameters are fixed. We remove ambiguity in Eq. 10 by using a semicolon to separate the optimized parameters from those held fixed.
> > Correct Generalized Advantage Extimation (GAE) formula in eq. 20
>
> Changed the GAE formula, now Eq.37, to the one in [1]
> ## Regarding W2
> > W2: Narrow set of baselines. More Async RL baselines.
>
> We agree and expanded the baseline set. The revised manuscript now includes an FRL baseline using FedProx[2] and MAPPO[3]. The FedProx experiments are reported in Tab. 14–17 (Sec. 6.5); FAuNO consistently **matches or outperforms FedProx, with clear gains in Finished Tasks Ratio.**
>
> For MAPPO, we use an oracle variant in which the critic observes the full global state. Since this requires bypassing network-level state propagation, MAPPO serves as an **upper bound on achievable performance**. The corresponding results are included in Sec. 6.7 (Tab.  22–25).
> ## Regarding W3
> > W3: Reward shaping limits generalizability of solution, and may require retuning across deployments.
>
> We acknowledge the concern. We added an alternative reward that retains only task-level utility and drop penalties. We then added a reward shaping mechanism that only used local node information. FAuNO remains stable under this formulation, though learning becomes harder as expected. We also note that in the random-topology setting LQ is effectively optimal due to the abundance of remote computation, serving as a practical performance ceiling. To strengthen the manuscript, we expanded the discussion of reward design (line 429) and added a new Sec. 6.6 with the alternative formulation and experiments.
> ## Regarding W4: Constraint vs. RL mismatch.
> > W4: Expliciting the enforcing and relaxation of optimization constraints into penalties
>
> We agree. In practice, the constraints are relaxed and incorporated as penalties in the reward. To clarify this, we revised line 238 to:
> “Although the constraints are enforced by the environment dynamics, in practice they are relaxed and incorporated as penalties in the optimization objective.”
> ## Regarding W5: Observation processing.
> > W5: Observation normalization and padding with −1 for missing neighbors.
>
> We agree this requires clarification. Our approach is effectively equivalent to padding with 1 and masking missing entries with −1. Zero-padding is unsuitable because 0 is a valid observation, even before normalization, and would be indistinguishable from real values. Using −1 provides a placeholder outside the feasible range to reliably signal missing neighbors.
> ## Regarding W6: Broken repository link
> > W6: Broken anonymous repository link.
>
> We have fixed the [link](https://tinyurl.com/5n6bamtz).
>
>
> ## Regarding Q4: Hierarchical and Decentralized Extension
> > Q4: Hierarchical or fully decentralized global critic.
>
> We agree this is an interesting and natural extension. As noted in the Future Work section (line ~489), hierarchical or decentralized aggregation can be built on top of the current formulation. However, these directions are feasible but beyond the scope of this work.
>
>
> ## Regarding Q5: Choice of $\epsilon$
> > Q5: Epsilon parameter choice
>
> The choice of $\epsilon$ was guided by empirical results in early development. During our experimentation using smaller $\epsilon$ produced results similar to those obtained with $\epsilon = 0.5$.
>
> ---
>
> We appreciate the reviewer’s feedback. The revisions directly address the issues raised and improve the clarity, completeness, and empirical support of the paper. We believe these changes strengthen the presentation and better convey the contributions of FAuNO.
>
>
> [1] J. Schulman, P. Moritz, S. Levine, M. Jordan, and P. Abbeel. High-dimensional continuous control using generalized advantage estimation. arXiv, 2018.
>
> [2] T. Li et al Federated Optimization in Heterogeneous Networks. Proc. MLSys, 2:429–450, 2020.
>
> [3] C. Yu et al. The Surprising Effectiveness of PPO in Cooperative Multi-Agent Games. NeurIPS 35:24611–24624.

---

### Official Review · Reviewer_UxDA · 2025-10-29

**Soundness:** 2
**Presentation:** 3
**Contribution:** 2
**Rating:** 4
**Confidence:** 3

**Summary:**

This paper addresses decentralized task offloading in edge computing, proposing FAuNO under conditions of limited observability and communication constraints. Specifically, each node trains its actors locally using PPO, while only performing federated semi-asynchronous buffering aggregation of critics across nodes, allowing faster nodes to participate more frequently and preventing slower nodes from being blocked. Empirical evaluation shows that FAuNO out performs or matches FRL and heuristic baselines in terms of task completion time and task completion.

**Strengths:**

1.FedBuff's "buffered semi-asynchronous" concept is introduced into federated reinforcement learning, and only the critic is federated, taking into account both personalization and sample efficiency. The engineering implementation is complete and open source.
2.The article also performs communication-level event simulation on PeersimGym, which more closely resembles real-world edge links.

**Weaknesses:**

1.Insufficient theoretical analysis. The paper models task offloading as POMG and proposes a framework of "actor local, critic federation, and semi-asynchronous buffer aggregation." However, it lacks convergence or upper bounds on error under semi-asynchronous and staleness conditions. It also fails to analyze the estimation bias of federated critics under non-IID or distribution drift conditions.
2.Ablation depth. Since the core claim hinges on critic-only federation + semi-async, include ablations for (i) federating both actor+critic, (ii) actor-only, and (iii) purely synchronous critic aggregation to isolate where gains originate.

**Questions:**

1.What happens if paper also federate the actor (or federate neither)? Does critic-only federation still dominate?

---

> ### Author Response · Authors · 2025-12-03
>
> We thank the reviewer for their feedback!
>
> Their comments motivated the addition of new ablations on the federated components of the actor–critic architecture and helped us identify future work directions concerning convergence analysis and error bounds in the non-trivial task-offloading setting. Below we address the reviewer’s weaknesses (W) and questions (Q). All modifications to the paper are shown in green.
>
> ## Regarding W1
>
> > W1: Insufficient theoretical analysis.
>
> We appreciate the reviewer’s interest in the theoretical properties of semi-asynchronous federated actor–critic methods. While convergence results exist for components such as PPO-Clip and for specific classes of federated optimization algorithms, extending these guarantees to the full FAuNO setting—combining partial observability, actor–critic structure, asynchronous buffering, non-IID workloads, and network-level delays—remains nontrivial, and to the best of our knowledge is not a common practice in the reinforcement learning task offloading area, and is outside the scope of the present work. Our focus in this paper is on establishing an empirically validated framework for decentralized task offloading in realistic edge conditions. We view a full theoretical treatment of staleness, bias under heterogeneity, and convergence under semi-asynchronous aggregation as important future work, and we have added text in that direction. We have included this interesting future work direction into our future work section in line 491.
>
> ## Regarding W2 and Q1: Ablation
>
> > W2 and Q1: Paper missing ablations on choice of federated models. And the dominance of the federated critic against other variants.
>
> We agree. We have completed the ablations involving (i) actor-only federation and (ii) joint actor–critic federation and (iii) the synchronous version of FAuNO. The corresponding problem formulations and results are reported in Section 6.4 (Tables 10–13). In the experiments FAuNO with a federated critic beats all other algorithms in term of throughput when considering the realistic Edge topolgoies. In the geometric netwo it beats or matches both (i) and (ii). FAuNO in geometric networks beats (iii) for all lambda higher than 0.5. In the $\lambda=0.5$ case, perhaps due to the simplicity of the problem due to the small task arrival rate and the lack of a need to do meaningfull task orchestration the synchronous version slightly outperforms FAuNO, losing this edge in the more complex problems with higher task arrival rates.
>
>
> We thank the reviewer for their feedback. Their comments guided the includion of new ablations on the federated components and the identification of interesting future directions such as convergence and error-bound analysis in the task-offloading setting. We believe these changes strengthen the paper and address the concerns raised.

---

### Official Review · Reviewer_kNGC · 2025-10-30

**Soundness:** 2
**Presentation:** 2
**Contribution:** 2
**Rating:** 4
**Confidence:** 4

**Summary:**

This paper propose FAuNO, a task offloading framework integrates buffered semi-asynchronous aggregation with PPO in edge systems. FAuNO enables agents to learn the task offloading policies under heterogeneous conditions. The experiments shows that FAuNO outperforms the heuristic and FRL-based baselines in terms of task completion time and task completion.

**Strengths:**

1.  This paper provides a detailed review in Background & Related Work.
2.  This paper presents a comprehensive process for building system model.

**Weaknesses:**

1.The description of the global component requires improvement. The relationship between Eq.11 and Eq.12 is unclear, and their connection to Figure 2 lacks detailed explanation.
2.The description in Fig. 2 is redundant and confusing, unable to convey the paper's idea.
3.The method proposed in this paper mostly relies on combining existing approaches (the PPO algorithm and FedBuff), which lacks innovation.
4.The paper has limited baselines for comparison, only one heuristic method and one FRL -based approach. The proposed method appears to be a straightforward FRL implementation. It is suggested to increase the number of FRL-based baselines and provide a more in-depth explanation of the advantages of the proposed method in the performance evaluation.
5.The specific definitions and descriptions of Variant in Tables 6 and 7 are unclear.

**Questions:**

1.What are the differences between Formula 11 and Formula 12, and which subgraphs in Figure 2 do they correspond to respectively?
2.What is the specific meaning of “Variant” in Tables 6 and 7? For example, FAuNO vs. FAuNO and Pure MARL vs. Pure MARL.

---

> ### Author Response · Authors · 2025-12-03
>
> We thank the reviewer for their thoughtful comments!
>
> During the rebuttal we used the reviewer’s comments to identify unclear or incomplete parts of the submission and revised it accordingly. The concerns raised were primarily about clarity, presentation, and insufficient detail on several components. We have addressed each point through targeted corrections, clarifications, and additional experiments. Below we address each raised weakness (W) and question (Q) and summarize the revisions. We highlight the changes in the paper with green.
> ## W1 and Q1: Global component, Eq.11/12, and Fig.2
> >W1: The description of the global component requires improvement.
> >Q1: Differences between Eq.11 and Eq.12, and relation to subgraphs of Fig.2?
>
> Eq.12 resulted from an unintended duplication of Eq.11. It has been removed, and Eq.11 is the sole expression governing aligning. Eq. 11 corresponds to Fig.2(d), which depicts the global training flow. To eliminate confusion, we revised line 316 (310 in the original) to:
> >Fig.2 provides an overview of the FAuNO global training process.
>
> The figure caption was rewritten to map each sub-figure to its corresponding algorithm stage.
> ## W2: Clarity of Fig.2
> > W2: The description in Fig.2 is redundant and confusing.
>
> We agree and revised Fig.2 accordingly. The new version includes (i) a concise caption, (ii) removal of redundant wording, and (iii) a clearer sequential description of the semi-asynchronous aggregation flow.
> Combined with Algo. 1, the figure now presents a clear description of the global update process.
> ## W3: Novelty of FAuNO
> >W3: The method combines existing approaches (PPO and FedBuff) and lacks innovation.
>
> We respectfully clarify that FAuNO is **not** a simple combination of PPO and FedBuff. Its novelty lies in being, to the best of our knowledge, the **first adaptation of FedBuff to reinforcement learning in a partially observable, federated RL setting,** which requires addressing challenges absent in supervised FL:
> - Continuous-training extension.
> FedBuff assumes clients stop training after sending an update; this is incompatible with RL, where discarding trajectories destabilizes learning. FAuNO introduces a continuous-training mechanism that preserves on-policy stability while allowing asynchronous aggregation.
> - Buffered semi-asynchronous critic-only federation.
> Critics are federated while actors remain local, enabling policy specialization under partial observability while sharing global value information. This architecture is algorithmically distinct from existing FRL approaches.
> - Straggler-robust asynchronous update weighting.
> FAuNO replaces stale critic updates on arrival and weights updates by local training steps, ensuring stable aggregation under heterogeneous device conditions.
> These adaptations constitute the core algorithmic contribution and are now clearly articulated in Sec.3 (revised for clarity).
> ## W4: Limited baselines
> > W4: Limited baselines; more FRL baselines and deeper comparison are needed.
>
> We agree and expanded the baseline set. The revised manuscript now includes an FRL baseline using FedProx[1]. New results (Tab. 14–17, Sec.6.5) evaluate FAuNO vs. FedProx in all geometric and edge topologies across load levels $\lambda \in \{0.5,1,2\}$.
>
> FAuNO consistently **matches or outperforms FedProx**, with **clear gains in Finished Tasks Ratio**.
>
> We also expanded the discussion in Sec.4, clarifying why FAuNO’s design improves sample efficiency and stability relative to centralized-aggregation FRL methods.
> ## W5 and Q2: Definition of "Variants"
> >W5: The definition of "Variant" in Tab. 6 and 7 is unclear.
> Q2: What is the meaning of FAuNO vs. FAuNO and Pure MARL vs. Pure MARL?
>
> We thank the reviewer for pointing out the ambiguity. Although the definitions were present in Sec.6.1, we agree that they were insufficiently referenced.
> In the revised manuscript:
> - Line 448 (446 previously) now explicitly refers readers to Sec.6.1 for variant definitions.
> - Lines 724–729 (670–674 previously) have been rewritten for clarity.
> The variants are now clearly defined as follows:
> - FAuNO: full method with semi-asynchronous global critic aggregation.
> - Pure MARL PPO: fully decentralized RL; agents train using local observations only, with no global critic (discard all incoming critic updates).
> - Centralized oracle: a non-decentralized upper bound where a single global critic is trained directly using all agents’ experiences.
> ___
>
> We appreciate the reviewer’s feedback. The revised paper addresses all raised concerns and significantly strengthens clarity, methodological soundness, and empirical rigor. We hope the improvements make the contributions of FAuNO clearer and demonstrate its relevance to FRL under realistic, heterogeneous edge-computing conditions.
> [1] T. Li et al. Federated Optimization in Heterogeneous Networks. Proc. MLSys, 2:429–450, 2020.
> [2] J. Nguyen et al. Federated learning with buffered asynchronous aggregation. CoRR, abs/2106.06639, 2021.

---

### Author Response · Authors · 2025-12-03

# FAuNO

We propose FAuNO, a semi-asynchronous federated reinforcement-learning framework for decentralized task offloading in edge systems.
Task-offloading in edge systems demands decentralized coordination under partial observability, dynamic workloads, and heterogeneous resources. Classical optimization and centralized controllers struggle with latency, network bottlenecks, and the inability to scale with distributed nodes. Multi-agent Reinforcement Learning is well-suited for such environments because agents can adapt to time-varying conditions and learn cooperative offloading strategies. Federated Reinforcement Learning (FRL) further enables coordination without global state visibility and without incurring the communication cost of sharing full experience, making it an appropriate paradigm for realistic edge deployments.

FAuNO uses an Actor-critic algorithm based on Proximal Policy Optimization [1] with with a federated critic updated through a buffered semi-asynchronous scheme. Each node trains the model locally using only local observations. The critic parameters are periodically shared with a global manager. This avoids relying on constructing a global state for decision making, keeping the execution fully decentralized. But still allowing for the agents to improve together.

The aggregation mechanism extends FedBuff [2] algorithm for continuous training. After completing a local training round, each agent sends its critic update to the global manager and continues training. If additional local steps occur before the previous update is aggregated, the new update replaces the old one in the buffer and receives a larger weight. Aggregation is triggered once updates from $K$ distinct agents are present. This prevents the idle time inherent to synchronous FRL and reduces sample waste under heterogeneous compute or communication speeds. After aggregation, the updated critic is sent back to all nodes and used in the next local training window.

FAuNO was implemented in an extended version of PeersimGym [3] that supports federated update exchange over the simulated network. Providing a setting consistent with practical edge deployments. The task-offloading problem is formulated as a partially observable Markov game, enabling decentralized learning under limited visibility. We evaluated FAuNO on realistic edge topologies and on random geometric networks, using Poisson task arrivals, realistic link-delay modeling, and workloads derived from computing-cluster traces. Baselines included Least-Queues (LQ) and the synchronous FRL method SCOF [4]. Across realistic topologies, FAuNO achieved the highest task-completion rates while maintaining competitive response times. In random networks, it outperformed SCOF and reduced response times relative to LQ. Under heterogeneous non-IID workloads, its federated critic remained consistent and approached a centralized oracle, whereas pure MARL diverged. Tests with stragglers and packet drops showed gradual degradation without collapse.

[1] John Schulman, Filip Wolski, Prafulla Dhariwal, Alec Radford, and Oleg Klimov. Proximal policy
optimization algorithms. 2017. doi: 10.48550/arXiv.1707.06347. arXiv:1707.06347 [cs].

[2] John Nguyen, Kshitiz Malik, Hongyuan Zhan, Ashkan Yousefpour, Michael G. Rabbat, Mani
Malek, and Dzmitry Huba. Federated learning with buffered asynchronous aggregation. CoRR,
abs/2106.06639, 2021

[3] Frederico Metelo, Cláudia Soares, Stevo Rackovi´c, and Pedro Ákos Costa. Peersimgym: An
environment for solving the task offloading problem with reinforcement learning. In Machine
Learning and Knowledge Discovery in Databases. Applied Data Science Track, pp. 38–54, Cham, 2024. Springer Nature Switzerland. ISBN 978-3-031-70378-2.

[4] Kai Peng, Peiyun Xiao, Shangguang Wang, and Victor C.M. Leung. Scof: Security-aware com-
putation offloading using federated reinforcement learning in industrial internet of things with
edge computing. IEEE Transactions on Services Computing, 17(4):1780–1792, 2024. doi:
10.1109/TSC.2024.3377899.

---

> ### Author Response · Authors · 2025-12-03
>
> ### Summary of modifications to the paper during the rebuttal (all highlighted in green)
>
> - Removed Eq. 12; clarified Eq. 11 and its mapping to Fig.2.
>
> - Rewrote Fig. 2 and corresponding text for clarity.
>
> - Clarified all variant definitions (FAuNO / Pure MARL / Oracle).
>
> - Provided clearer exposition of novelty: continuous-training extension of FedBuff, critic-only federation, robust asynchronous update weighting.
>
> - Fully updated anonymous repository with all code and instructions.
>
> - Added discussion on theoretical limitations and included convergence/error-bound analysis as future work (line 491).
>
> - Added ablations on federated components: actor-only, joint actor–critic, and synchronous variants (Sec. 6.4; Tables 10–13).
>
> - Strengthened and explicitly highlighted the single–point-of-failure limitation (revised line ~482).
>
> - Further clarified that hierarchical or decentralized critics are natural extensions in the Future Work (~line 485)
>
> - Expanded Sec. 10.3 to clarify the realism of the simulation environment (Alibaba traces[4]  + Ether topologies[5]).
>
> - Clarified in Limitations that FAuNO assumes cooperative, non-adversarial settings and does not address poisoned updates.
>
> - Clarified and corrected equations:
>   - Updated Eq. 1 to explicitly use log₂.
>   - Corrected the sign of the delay term in Eq. 9 to match the implementation.
>   - Clarified fixed actor parameters in Eq. 10 using a semicolon.
>   - Corrected the GAE formula in Eq. 20 following [1].
>
> - Expanded baselines and empirical coverage:
>   - Added FedProx FRL baseline [1] (Tables 14–17, Sec. 6.5).
>   - Added MAPPO oracle baseline[3] (Tables 22–25, Sec. 6.7) as an upper-bound reference.
>
> - Added alternative reward formulation:
>   - Introduced a sparse reward using only task utility and drop penalties.
>   - Added local-only shaping variant; discussed the effects on learning (Sec. 6.6).
>
> - Clarified handling of optimization constraints:
>   - Revised line 238 to note that constraints are relaxed and implemented as penalties.
>
> - Clarified observation normalization:
>   - Explained the use of −1 as a mask to avoid collisions with valid observations.
>
> - Fixed broken anonymous repository link.
>
> - Added discussion on extending the critic:
>   - Addressed hierarchical and decentralized critic designs (Future Work, line 489).
>
> - Clarified epsilon choice:
>   - Explained empirical motivation and noted that smaller $\epsilon$ produced comparable results.
>
> We thank the AC for the time and effort dedicated to reviewing the submission. This synthesis summarizes the revisions and additions made during the rebuttal period.
>
> [1] T. Li et al Federated Optimization in Heterogeneous Networks. Proc. MLSys, 2:429–450, 2020.
>
> [2] J. Nguyen, K. Malik, H. Zhan, A. Yousefpour, M. Rabbat, M. Malek, and D. Huba. Federated learning with buffered asynchronous aggregation. CoRR, abs/2106.06639, 2021.
>
> [3] C. Yu et al. The Surprising Effectiveness of PPO in Cooperative Multi-Agent Games. NeurIPS 35:24611–24624.
>
> [4] Huangshi Tian, Yunchuan Zheng, and Wei Wang. Characterizing and synthesizing task dependencies of data- parallel jobs in alibaba cloud. In Proc. ACM Symp. Cloud Comput., 2019.
>
> [5] Thomas Rausch, Clemens Lachner, Pantelis A Frangoudis, Philipp Raith, and Schahram Dustdar. Synthesizing plausible infrastructure configurations for evaluating edge computing systems. In 3rd USENIX Workshop HotEdge 20, 2020.

---

### Meta-Review · Area_Chair_2TZn · 2025-12-03

**Summary:**

Based on the reviewers’ feedback and my own reading of the paper, the overall quality still needs improvement. We regret to inform you that this paper has not been accepted for this year’s conference. We hope the authors can address the relevant issues in subsequent revisions and achieve acceptance in future submissions.

**Reviewer Concerns:**

No rebuttal.

**Reviewer Scores:**

The paper has limited baselines for comparison, only one heuristic method, and one FRL -based approach.

---

### Decision · Program_Chairs · 2026-01-26

Reject